# SUBSPACE NODE PRUNING: REVEALING STRUCTURED IMPORTANCE VIA ORTHOGONAL SUBSPACE TRANSFORMATIONS

## ABSTRACT

Improving the efficiency of neural network inference is undeniably important in a time where commercial use of AI models increases daily. Node pruning is the art of removing computational units such as neurons, filters, attention heads, or even entire layers to significantly reduce inference time while retaining network performance. In this work, we propose the projection of unit activations to an orthogonal subspace in which there is no redundant activity and within which we may prune nodes while simultaneously recovering the impact of lost units via linear least squares. We furthermore show that the order in which units are orthogonalized can be optimized to maximally rank units by their redundancy. Finally, we leverage these orthogonal subspaces to automatically determine layer-wise pruning ratios based upon the relative scale of node activations in our subspace, equivalent to cumulative variance. Our method matches or exceeds state-of-the-art pruning results on ImageNet-trained VGG-16, ResNet-50 and DeiT models while simultaneously having up to $24\times$ lower computational cost than alternative methods. We also demonstrate that this method can be applied in a one-shot manner to OPT LLM models, again outperforming competing methods.

## 1 INTRODUCTION

A variety of approaches have been developed to reduce the computational footprint of evermore computationally expensive neural network models. These range from low-level hardware optimizations (Jouppi et al., 2018; Choquette et al., 2021) to high-level software developments (Abadi et al., 2015; Bradbury et al., 2018; Paszke et al., 2019). Additionally, the representations of models in software have been made more compact with quantization methods (Krishnamoorthi, 2018; Gholami et al., 2022). More promising, however, are pruning methods which modify and compress neural network models to reduce computational cost while maintaining accuracy.

**Pruning neural networks** The goal of neural network pruning is to reduce the computational execution (inference) time of a model while maintaining its performance. Unstructured approaches prune the weights of a model, resulting in arbitrarily sparse weight matrices whose multiplication cannot easily be accelerated at compute time, i.e., without translation to real-world inference efficiency. It is therefore desirable to prune whole nodes, convolutional filters, transformer heads, or other structured groups of parameters. Herein, we refer to any of these sub-parts of networks as network 'units'. Two broad approaches have emerged to attempt network pruning. First, pruning of pre-trained networks and, second, pruning iteratively while training networks. While we limit our investigation to the former class that assumes starting out with a well-performing model, the latter class holds great potential for also reducing the cost of training next to the cost of inference.

**Importance scores** When choosing which units of a network to prune, there must first be an attribution of the relative importance of each network unit. A number of methods use the magnitude of weights as a proxy for their importance score, assuming that smaller weights can be removed without impacting a network's computation (Li et al., 2016). Methods of greater complexity are 'data-driven', making use of training data for forward, and in some cases backward, passes enhance importance measures. These use ideas such as the $1^{\text{st}}$-order (or $2^{\text{nd}}$-order) Taylor-expansion of a

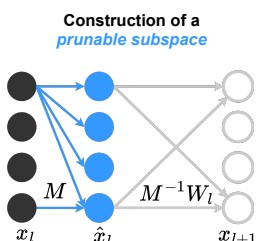 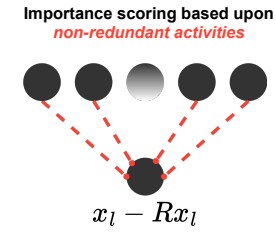 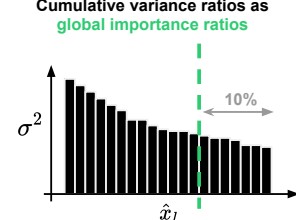

Figure 1: Graphical depiction of our three main method and contributions. From left to right: The construction of a subspace in which nodes can be pruned with automated reconstruction, a theoretically sound importance scoring method which aligns with our subspace construction, and finally an automated method based upon cumulative variance for selecting automatically selecting pruning ratios for all layers of a network.

network's input to output mapping (Molchanov et al., 2016; 2019), Fisher information (Theis et al., 2018), or even downstream network importances to compound importance measures (Yu et al., 2018).

Other approaches, which do not necessarily consider small weights as unimportant, look at feature maps to determine filters which have greatest task-relevant information (Liu et al., 2023; Zhang et al., 2022a) or various correlation measures which equate high correlation between units as indicating low importance (Mariet & Sra, 2015; Ayinde et al., 2019; Cuadros et al., 2020; Kim et al., 2020; Goldberg et al., 2022; Zhang et al., 2022a; He et al., 2019).

**One-step reconstruction** Some methods go beyond the step of simply removing nodes when pruning and additionally carry out a form of 'one-step reconstruction'. This reconstruction modifies the left-over parameters of a pruned network to undo the effect of having removed parameters or nodes. This is unlike retraining or finetuning as it is a single step of modification.

Mariet & Sra (2015) identified redundant nodes by using determinantal point processes, pruned these nodes, and recovered their impact on a network by linear least squares (LLS). Similarly, He et al. (2017) used LLS to approximate pruned nodes, while selecting nodes based on LASSO regression. Other work uses less expressive approaches, with single scalar values used for reconstruction per node (Luo et al., 2017), or even employ data-free approaches to reconstructing unit activity (Theus et al., 2024). Other work goes further still, into a non-linear least squares solution attempt using evolutionary algorithms (Chin et al., 2018), or application of approximate reconstruction even to large language models (Frantar & Alistarh, 2023; Li et al., 2024).

**Global importance** Despite the importance of local scoring (i.e. scoring of units within a layer), there needs to be a notion of unit importance across layers, a global calibration of importance scores. While for some approaches such global ranking is a natural consequence of the local importance estimation (Molchanov et al., 2016; 2019; Yu et al., 2018), several methods only provided local importances and rely on expert knowledge, manual exploration (Wang et al., 2021; Wang & Fu, 2023), or simple assumptions such as the equivalence of pruning at any layer (Li et al., 2016). The most advanced methods rely on the measurement of some form of network 'sensitivity' to achieve peak performance (You et al., 2019).

**Pruning Transformers** Methods in transformer model pruning go beyond estimating local or global importance, often pruning different granularities of nodes simultaneously. Zheng et al. (2022) designed an optimization problem to prune attention heads, embedding dimensions and the hidden layers of the feedforward networks. They combined a 2nd-order Taylor estimation with evolutionary algorithms to optimize a pruning mask. Yang et al. (2023b) went even further by pruning *query* and *key* embedding dimensions independent of the *value* embedding dimension. These and several other recent but influential methods train for estimating global sparsity (Yu & Xiang, 2023; Yang et al., 2023a; Yu et al., 2022). Dynamic removal of tokens/patches to reduce the inference time for a given sample is also an area under increasing investigation (Rao et al., 2021; Song et al., 2022; Wang et al., 2022; Liang et al., 2022).

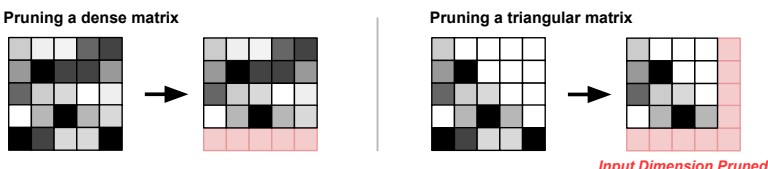

Figure 2: Our choice of a subspace which is constructed for lower-triangular matrices is here justified. Left: If a dense matrix is used to form a subspace, pruning does not prune the original input nodes. Right: When pruning a lower-triangular transformation matrix, pruning the bottom row corresponds to pruning away an entire input node.

**This work** Herein we contribute a novel perspective on one-step reconstruction, which serves as a foundation for two additional new methods for estimating unit importance. Specifically, we contribute:

1. A novel approach to node pruning via a subspace in which unit activations are factorized and, when pruned, any contribution from lost nodes is automatically reconstructed via linear least squares.

2. An efficient node-level importance scoring method based upon how redundant the activity of a unit is with respect to all other units in a layer.

3. A global importance metric based upon the percent of variance explained within our proposed subspace, allowing automated selection of pruning ratios across an entire network.

Figure 1 illustrates these three contributions in order, and these are described in detail in the sections which follow. Further, we demonstrate the effectiveness and computational efficiency of our approach in comparison to the state-of-the-art by application to ImageNet-trained VGG-16, ResNet-50 and DeiT models, as well as trained OPT language models tested with the WikiText dataset.

## 2  SUBSPACE NODE PRUNING

Consider a typical deep neural network (DNN) architecture, in which the outputs at each layer, $l \in \{1, \ldots, L\}$, are defined as $\mathbf{X}_l = f_l(\mathbf{Y}_l) = f_l(\mathbf{W}_{l-1}\mathbf{X}_{l-1})$, where $\mathbf{X}_l \in \mathbb{R}^{n_l \times s}$ is a tensor of outputs for layer $l$ consisting of $n_l$ units for $s$ samples. These layer outputs are composed based upon a matrix multiplication of the previous layer outputs, from weights $\mathbf{W}_l \in \mathbb{R}^{n_{l+1} \times n_l}$, and by an element-wise transfer function $f_l(\cdot)$. We consider these fully-connected deep neural networks to introduce our approach, however, we also apply this theory to convolutional and transformer architectures in the results that follow.

### 2.1  FACTORIZING NEURAL CONTRIBUTIONS

Assuming that importance scores are already available, a pruning pipeline would next prune $m$ input units and their associated weight vectors, starting with the unit of lowest importance score.

In this section we propose that, prior to pruning, one may linearly transform the input data to a basis in which all redundant information has been removed from the units to be pruned. Subsequent pruning within such a basis lets us target unique unit activities only, whereas redundant activities are untouched to minimize shifts in the expected activity in downstream computations.

**Definition 1:** Let $\mathbf{M}_l$ be a transformation matrix which transforms our input data to a subspace, $\hat{\mathbf{X}}_l = \mathbf{M}_l \mathbf{X}_l$, where the unit activities are now orthogonalized (statistically uncorrelated), such that

$$\hat{\mathbf{X}}_l \hat{\mathbf{X}}_l^\top = \mathbf{M}_l \mathbf{X}_l \mathbf{X}_l^\top \mathbf{M}_l^\top := \mathbf{D}_l,$$

where $\mathbf{D}_l$ is a diagonal matrix of the remaining variances of the unit activities within the subspace, our latent variables. Note that, assuming $\mathbf{M}_l$ is invertible, the network's output can remain unchanged when $\mathbf{Y}_{l+1} = \mathbf{W}_l \mathbf{X}_l = \mathbf{W}_l \mathbf{M}_l^{-1} \hat{\mathbf{X}}_l$

Orthogonalization is possible in multiple ways, for example, by principal component analysis (PCA), zero-phase component analysis (ZCA) or otherwise. However, pruning in these subspaces does not result in a reduced matrix dimensionality, i.e. does not prune any nodes, but instead in a low-rank matrix (Levin et al., 1993). Therefore, we ensure that our orthogonalizing transformation matrix, $\mathbf{M}_l$, is lower-triangular. In order to understand why, consider two aspects. First, a lower-triangular orthogonalizing matrix means that our units are treated as if they have been ordered by priority, with the first unit orthogonalizing the $n_l - 1$ remaining units, the second unit orthogonalizing the $n_l - 2$ remaining units and so on. This ensures that the final units of the layer (the units which will be pruned first) have had all possible activity which could be explained by earlier units removed. Second, this lower-triangular setup allows one to prune the latent variables whilst also pruning the original input nodes. This prunability is a natural consequence of the zeros in the upper-triangular section of our transformation matrix, as illustrated in Figure 2, that lead to removing columns through pruning rows from $\mathbf{M}_l$.

Given this constraint, one can show that LDL decomposition provides the desired lower-triangular orthogonalizing transformation matrix, $\mathbf{M}_l$. This can be demonstrated by considering the re-arrangement of the above equation such that

$$\mathbf{X}_l \mathbf{X}_l^\top = \mathbf{M}_l^{-1} \mathbf{D}_l \left( \mathbf{M}_l^{-1} \right)^\top .$$

Supposing that $\mathbf{M}_l$, and therefore $\mathbf{M}_l^{-1}$, is lower-triangular, this precisely describes that our desired transformation matrix can be obtained by an LDL decomposition of the Gram matrix of our input data. Note that this transformation is equivalent to an unnormalized Gram-Schmidt (GS) orthogonalization of the input data, a well-known method for orthogonalizing vectors.

In this proposed subspace, we can now prune from most- to least 'restricted' latent variable (restricted by the triangular structure of the matrix $\mathbf{M}_l$). Therefore, we prune $\mathbf{M}_l$ by removing rows from the bottom. If we denote $\mathbf{M}_{l,(*,)}$ as pruning the last rows, and $\mathbf{M}_{l,(,*)}$ as pruning the last columns of a matrix $\mathbf{M}$ of layer $l$, we reparameterize the weights as

$$\hat{\mathbf{W}}_l = \mathbf{W}_l (\mathbf{M}_l^{-1})_{(,*)} \mathbf{M}_{l,(*,*)} .$$

The full pipeline of factorization and pruning is described in Algorithm 1. Note that a similar algorithm can be used to prune entire filters in convolutional networks.

---

**Algorithm 1:** Layer-wise subspace node pruning

**Input:** Data $\mathbf{X}_l$, Weights $\mathbf{W}_l$, Number of units to prune $n$
**Output:** Pruned weights $\hat{\mathbf{W}}_l$
$\mathbf{C}_l = \mathbf{X}_l \mathbf{X}_l^T$              ▷ Compute dot-product between input feature vectors
$\mathbf{M}_l^{-1}, \mathbf{D}_l = \mathrm{LDL}(\mathbf{C}_l)$                    ▷ Decompose matrix $\mathbf{C}$
$\hat{\mathbf{W}}_l = \mathbf{W}_l \mathbf{M}_{l,(:,:n)}^{-1} \mathbf{M}_{l,(:n,:n)}$       ▷ Prune $\mathbf{M}$ and $\mathbf{M}^{-1}$ (leading to pruned $\mathbf{W}$)
**Return:** $\hat{\mathbf{W}}_l$

---

In Appendix A, we prove that pruning in this subspace automatically reconstructs the output of such a layer by LLS approximation. This equivalence demonstrates the optimality of our choice of subspace. It highlights that via LLS one recovers all input activity that was redundant, i.e. activity which could also have been read out from units that remain in the network, and that the only activity pruned is unique to the pruned nodes.

## 2.2 IMPORTANCE SCORING: REORDERING UNITS PRIOR TO FACTORIZATION

So far we constructed a subspace transformation assuming that input units are pruned based upon their 'default' ordering, an ordering which can be much improved. Generally, the choice of unit ordering is free for a practitioner since it simply changes the order of units from which we compute the GS subspace (consider that one could permute the matrix $\mathbf{M}_l$ so long as you also unpermute via matrix $\mathbf{M}_l^{-1}$). See Appendix B for the pseudo-code of this permutation for pruning individual layers in a network.

Here, we propose that unit importances are best identified within an orthgonal subspace, such that we only estimate the importance of the non-redundant unit activities. This is equivalent to reordering the units in a layer prior to the factorization step to maximize the utility of said factorization.

To determine the ideal ordering of units in a layer, one would be required to check every possible permutation. This is prohibitively expensive and defeats the purpose of efficient methods for importance scoring. Instead we propose to align with our previous factorization method by asking: If we did not have a triangular restriction upon the orthogonalization method, which order would maximally minimize the subspace variances?

Following Definition 1, let $\mathbf{R}_l$ be a symmetric subspace transformation matrix with a diagonal of 1s, such that $\bar{\mathbf{D}}_l = \bar{\mathbf{X}}_l \bar{\mathbf{X}}_l^\top$, where $\bar{\mathbf{X}}_l = \mathbf{R}_l \mathbf{X}_l$. Then, our question becomes for which $\mathbf{R}_l$ is each value in $\bar{D}_l$ minimized?

Note that the diagonal matrix, $\bar{\mathbf{D}}_l$, would in fact be our desired set of importance scores as they capture the variance remaining in any unit when orthogonalized by all others. Also note that solving this problem includes the restriction that the diagonal of $\mathbf{R}_l$ must be 1s, otherwise trivial solutions appear where this is simply a matrix of zeros.

To solve for our diagonal matrix $\bar{\mathbf{D}}_l$, we use the definition that

$$\bar{\mathbf{X}}_l \bar{\mathbf{X}}_l^\top = \mathbf{R}_l \mathbf{X}_l \mathbf{X}_l^\top \mathbf{R}_l^\top = \bar{\mathbf{D}}_l \,,$$

which, by rearrangement means that $\mathbf{X}_l \mathbf{X}_l^\top = \mathbf{R}_l^{-1} \bar{\mathbf{D}}_l \mathbf{R}_l^{\top^{-1}}$. We can further take the square root of our diagonal matrix such that $\mathbf{S}_l^2 = \bar{\mathbf{D}}_l$, giving

$$\mathbf{X}_l \mathbf{X}_l^\top = \mathbf{R}_l^{-1} \mathbf{S}_l^2 \mathbf{R}_l^{\top^{-1}}.$$

By assuming that our transformation is symmetric there is a trivial solution by taking the matrix square root, such that $\mathbf{S}_l^{-1} \mathbf{R}_l^\top = (\mathbf{X}_l \mathbf{X}_l^\top)^{-1/2}$. This transformation is equivalent to the well known ZCA transform (Krizhevsky et al., 2009), although we have now added a term which explicitly represents the scaling that is applied to reach a whitened state.

With this form, as well as the prior constraint that the diagonal of $\mathbf{R}_l^\top$ is composed of 1s, it is easy to note that the diagonal of this term must be equivalent to the diagonal of our matrix $\mathbf{S}_l$ such that

$$\mathrm{diag}(\mathbf{S}_l) = \mathrm{diag}((\mathbf{X}_l \mathbf{X}_l^\top)^{-1/2})^{-1} \,.$$

The (diagonal) values of the matrix $\bar{\mathbf{D}}_l$ ($= \mathbf{S}_l^2$) are the novel importance scores which we propose in this work and we refer to this as the 'unnormalized-ZCA' ordering. Concretely, these values are the L2-norms of each units' activation after each unit has individually been orthogonalized by all other units. This effectively means that it is the scale of each units' activation which is truly unique (non-redundant) with respect to all other unit activations. See Appendix C for a relation of this importance measure to existing measures.

It is also possible to generalize the idea and to combine this measure with other existing importance scoring methods in order to discount redundant information when measuring importance. We briefly describe such an extension in Appendix D but do not explore it any further in this work.

### 2.3 CUMULATIVE VARIANCES: FROM PRUNING LAYERS TO PRUNING NETWORKS

In the previous section, we described the measurement of an importance score based upon the remaining norm of a unit's activity after it has been orthogonalized by all other units within a layer. Although these importance scores are an efficient one-shot estimation of layer-wise unit orderings, these scores assume that each unit can continue to contribute to the orthogonalization after being pruned. With local orderings available, we can use them to recompute the importance scores to correct for the loss of orthogonalization to higher ranked units during pruning. We use these corrected layer-wise importance scores to compare the importances across layers for a global aggregated unit ranking.

To this end, we notice that our subspace pruning method already yields the activity variances of the latent variables (subspace units) without extra computation (the diagonal matrix $\mathbf{D}_l$ from LDL-decomposition). To ensure that the variances reflect only the effect of orthogonalization, we restrict the diagonal of the orthogonalizing matrix $\mathbf{M}_l$ to be 1. With these variances computed, we propose to measure the global importance as the cumulative variance of a unit and all succeeding units,

normalized by the total variance of the layer. Mathematically speaking, this means that the global variance-based importance score for a given unit $i$ in layer $l$ is given by

$$\text{Importance}_l^i = \frac{\sum_{j=i}^{n_l} \mathbf{D}_{l,(j,j)}}{\sum_{k=0}^{n_l} \mathbf{D}_{l,(k,k)}} \,,$$

where the repeated subscripts indicate selection of diagonal elements of our $\boldsymbol{D}_l$ matrices. This construction allows us to set a single global parameter (the percent variance to be removed from all layers) which automatically arrives at an individual layer-wise pruning ratio.

## 3  EXPERIMENTS

**Networks and datasets**   To demonstrate the efficacy of our proposed method, we first apply it to single-branch models VGG-11, 16, and 19 (Simonyan & Zisserman, 2014), followed by application to ResNet-50 (He et al., 2016), and lastly the transformer architectures DeiT (Touvron et al., 2021) and OPT (Zhang et al., 2022b). OPT is evaluated on the WikiText dataset (Merity et al., 2016). All other models are pretrained and evaluated on the ILSVRC 2012 (ImageNet) dataset (Russakovsky et al., 2015) We refer to Appendix E for further details on this matter, and Appendix F for details on how we deal with pruning multi-branch networks. All code is available at <SEE ATTACHED ZIP>.

**Metrics and hardware**   We measure performance as the Top-1 test accuracy for vision models and perplexity for language models in relation to their FLOPs, parameter count, and wallclock time. We use the fvcore package (https://github.com/facebookresearch/fvcore) to estimate the FLOPs on the convolutional networks, and the dependency graph package (Fang et al., 2023) for DeiT models. All experiments were run on one Nvidia A100 GPU with 20 AMD Epyc 9334 CPU cores.

## 4  RESULTS

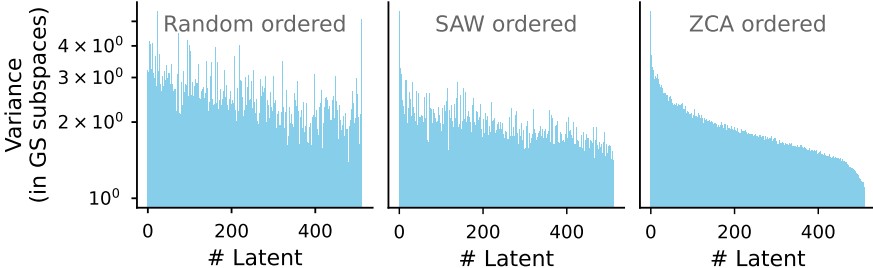

Figure 3: Latent unit variances in our subspace after Gram-Schmidt orthogonalization of layer 12 from VGG-16. Prior to the orthogonalization, the units are ordered either randomly (left), ordered using the SAW importance measure (middle) and ordered using our proposed ordering by unnormalized-ZCA variances (right).

First, we assess the efficacy of our proposed novel importance scoring method. Figure 3 shows the variances of the network units after they have been projected to our subspace, i.e. orthogonalized by our (unnormalized) Gram-Schmidt method. Permuting the unit order according to the importance scores of summed-absolute weights (SAW) (Li et al., 2016) (prior to GS orthogonalization) improves upon the case in which units are ordered randomly. This indicates that the choice of unit ordering given by the SAW method does measurably help to order units by redundancy of their activities, but not fully. In contrast, re-ordering according to our unnormalized-ZCA importance scoring method leads to a smooth and orderly ranking of variances showing that units are clearly ordered, left to right, from least to most activity redundant. This demonstrates the efficacy of our combined subspace and importance scoring methods.

**Single-branch convolutional networks**   Figure 4 (left) shows the post-pruning but pre-retraining accuracies of VGG-16 using our proposed method alongside various baselines. See Appendix E

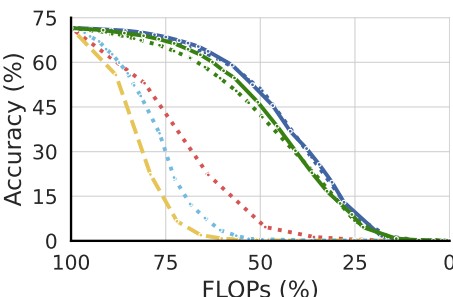 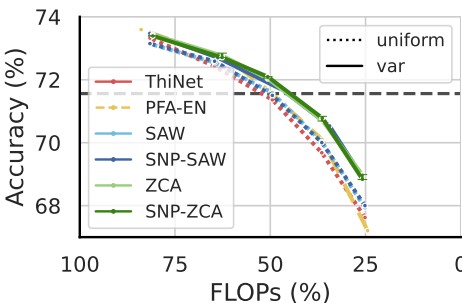

Figure 4: **Left:** Performance prior to retraining; **Right:** after retraining. A comparison of our subspace pruning (SNP) with our local ZCA-based importance (ZCA) and global variance cutoff (var) vs baseline methods on VGG-16. See Appendix E for details on the baseline methods. The black strided horizontal line (right panel) shows the initial network performance before pruning. SNP-ZCA has error bars on top of the datapoints from three randomly seeded training runs, though these are barely distinguishable. PFA-EN is the only unique method which uses PCA to determine global importance, indicated by the dashed line.

for details on the baseline methods. In the plot, we also compare our proposed global variance-based pruning against uniform pruning. As can be observed, the subspace methods (SNP-SAW and SNP-ZCA) are by far the most performant, retaining much of the initial performance with a slow degradation at greater pruning levels. Alternative methods suffer to a much greater degree with significant reductions in test accuracy even at small FLOP reductions. As yet in this case, the use of our cumulative variance-based pruning ratio selection is not necessarily more effective than a uniform ratio for all layers. These observations also hold for VGG-11 and 19 shown in Appendix G.

Figure 4 (right) shows the performance comparison after retraining. See Appendix H for a table of the exact final accuracies. Most crucially, our global variance-based cutoff for automated network-wide pruning shows significant performance gains over uniform pruning for FLOP reductions of $2\times$ and greater, demonstrating its efficacy and suitability when allowing networks to retrain. Our SNP-ZCA and SNP-SAW are very similar in performance and outperform all other methods by an increasing margin as we decrease the number of FLOPs.

The initial high performance of PFA-EN (Cuadros et al., 2020) is competitive with our SNP-ZCA and SNP-SAW for the first two pruning ratios, but then suffers from a stark drop-off. ThiNet (Luo et al., 2017) is largely outperformed by the other methods. Included is also an example of our proposed ZCA-based importance scoring without subspace reconstruction (see ZCA var), as well as the SAW baseline with subspace reconstruction (see SNP-SAW uniform). These results show that our reconstruction provides major improvement prior to retraining and a rather small impact when retraining is done to convergence. However, this does not yet capture any differences in the time taken to reach a converged state.

**Multi-branch convolutional (aka residual) networks** In Figure 5 we show that before retraining, our method massively outperforms Intra-Fusion (IF) (Theus et al., 2024) across pruning groups of a ResNet-50 model (see Appendix F for the definition of a pruning group). When we limit the estimation of the Gram matrix to 1024 images in our method, it still outperforms IF. Even when injecting white noise for its estimation, our method retains performance significantly longer than IF, at the cost of being outperformed by a small margin for compression ratios below 60%.

When we allow the network to recover via retraining, we obtain the results reported in Table 1. As can be seen, our method beats all methods with regular retraining by a wide margin. Notably, it also beats GReg-2 (Wang et al., 2021) and is highly competitive to TPP (Wang & Fu, 2023). It must be noted that our method is $24.3\times$ faster than TPP with a marginal top-1 accuracy loss, and unlike all other competitors, does not rely on manually selected pruning ratios for each and every layer. Note, that both TPP and our method require approximately 30 hours of retraining time, with TPP requiring an additional 12 hours of prior iterative pruning where our method requires only 30mins of prior pruning time. Appendix J, provides additional results for ResNet-56 applied to the CIFAR-10 task.

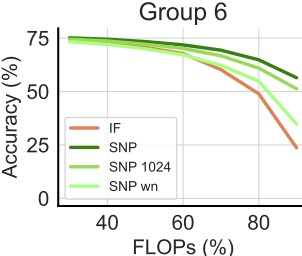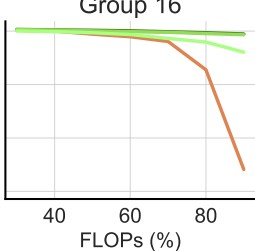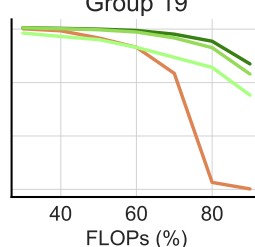

Figure 5: A comparison of pruning different groups of layers of ResNet-50. We compare our SNP-ZCA method with the variance heuristic against Intra-Fusion (IF). Further, we demonstrate our method when only using 1024 samples (SNP 1024) or white noise (SNP wn) is input to the network while constructing the Gram matrices.

Table 1: **ResNet-50 (ImageNet)**: The final accuracies, FLOP speed-up, and parameter count for our method along with referenced alternatives from the literature. *Pruning Cost* indicates the excess time required for this pruning method as a percent of training time. To ensure a fair comparison of our results and those in the literature, we group the results into blocks with similar FLOP speed-ups. Groups are separated by dashed lines. In blue we highlight our full proposed model.

| Importance | Pre-prune %Acc | Final %Acc | $\Delta$Acc | FLOP speedup | #Params | Pruning Cost (% Train Time) |
|---|---|---|---|---|---|---|
| Taylor-FO (Molchanov et al., 2019) | 76.18 | 74.50 | -1.68 | 1.82× | 14.2M | - |
| GReg-2 (Wang et al., 2021) | 76.13 | 75.36 | -0.77 | 2.31× | - | - |
| TPP(Wang & Fu, 2023) | 76.13 | **75.60** | **-0.53** | 2.31× | - | 41.38% |
| SAW (Wang et al., 2023) | 76.13 | 75.24 | -0.89 | 2.31× | - | **<0.1%** |
| FPGM(He et al., 2019) | 76.15 | 74.13 | -2.02 | 2.13× | - | - |
| **SNP-SAW var** (ours) | 76.13 | 75.08 | -1.05 | 2.30× | 11.16M | 1.7% |
| **SNP-ZCA uniform** (ours) | 76.13 | 75.18 | -0.95 | 2.34× | 11.24M | 1.7% |
| **SNP-ZCA var** (ours) | 76.13 | **75.43** | **-0.70** | 2.30× | 13.76M | 1.7% |
| LFPC (He et al., 2020) | 76.15 | 74.46 | -1.69 | 2.55× | - | - |
| GReg-2 (Wang et al., 2021) | 76.13 | 74.93 | -1.20 | 2.56× | - | - |
| TPP (Wang & Fu, 2023) | 76.13 | **75.12** | **-1.01** | 2.61× | - | 41.38% |
| SAW (Wang et al., 2023) | 76.13 | 74.77 | -1.36 | 2.56× | - | **<0.1%** |
| **SNP-SAW var** (ours) | 76.13 | 74.47 | -1.66 | 2.61× | 9.89M | 1.7% |
| **SNP-ZCA uniform** (ours) | 76.13 | 74.67 | -1.46 | 2.63× | 10.00M | 1.7% |
| **SNP-ZCA var** (ours) | 76.13 | **75.08** | **-1.05** | 2.60× | 12.37M | 1.7% |
| Taylor-FO (Molchanov et al., 2019) | 76.18 | 71.69 | -4.49 | 3.05× | 7.9M | - |
| GReg-2 (Wang et al., 2021) | 76.13 | 73.90 | -2.23 | 3.06× | - | - |
| TPP (Wang & Fu, 2023) | 76.13 | **74.51** | **-1.62** | 3.06× | - | 41.38% |
| SAW (reimpl.) | 76.13 | 74.13 | -2.00 | 3.03× | 8.77M | **<0.1%** |
| **SNP-SAW var** (ours) | 76.13 | 73.85 | -2.28 | 3.09× | 8.38M | 1.7% |
| **SNP-ZCA uniform** (ours) | 76.13 | 74.36 | -1.77 | 3.03× | 8.77M | 1.7% |
| **SNP-ZCA var** (ours) | 76.13 | **74.43** | **-1.70** | 3.04× | 10.74M | 1.7% |

**Transformer networks** Table 2 summarizes the retrained performance of our method on DeiT models. On DeiT-Tiny, our approach outperforms the leading baseline by nearly 0.8%, a substantial margin, and on DeiT-Small, we achieve competitive performance. Beyond accuracy, our method offers several practical advantages. Most notably, it is a simple, interpretable heuristic that estimates node importance without requiring any additional training for the pruning process. In contrast, competing approaches such as SAViT and S²ViTE involve training pruning-specific parameters, increasing computational cost and opacity of the method. Furthermore, our method operates on the full dataset but remains efficient. SAViT, in comparison, relies on only 10% of the data during pruning. In Appendix I, we demonstrate that our approach maintains its performance even when computed on a single mini-batch. This suggests that pruning can be performed in seconds rather than minutes, offering a substantial speedup. Appendix L further demonstrates that the FLOP reductions described here and above truly translate to practical inference time speedups. Finally, Appendix K

demonstrates that node pruning itself is also a far more practically efficient pruning strategy than the conceptually similar subspace pruning strategy of low-rank pruning.

Table 2: **DeiT (ImageNet)** Interpretation as in Table 1. All pruning times are measured on a the same hardware (Nvidia A100 GPU).

| | Importance | Final %Acc | FLOP speedup | #Params | Pruning time (min) | Dataset usage(%) |
|---|---|---|---|---|---|---|
| **DeiT-Tiny** (72.20%) | VBP (Berisha et al., 2025) | 70.64 | 1.38× | 3.94M | – | – |
| | SAW (Berisha et al., 2025) | 68.49 | 1.38× | 3.94M | – | 0 |
| | SSP (Chen et al., 2021) | 68.59 | 1.31× | 4.2M | – | 100 |
| | S²ViTE (Chen et al., 2021) | 70.12 | 1.31× | 4.2M | – | 100 |
| | SAViT (Zheng et al., 2022) | 71.08 | 1.33× | 4.2M | 23 | 10 |
| | **SNP-ZCA var (ours)** | **71.86** | 1.33× | 4.1M | **23** | 100 |
| **DeiT-Small** (79.85%) | VBP (Berisha et al., 2025) | 78.62 | 1.43× | 14.96M | – | – |
| | SAW (Berisha et al., 2025) | 76.55 | 1.43× | 14.96M | – | 0 |
| | SSP (Chen et al., 2021) | 77.74 | 1.46× | 14.6M | – | 100 |
| | S²ViTE (Chen et al., 2021) | 79.22 | 1.46× | 14.6M | – | 100 |
| | SAViT (Zheng et al., 2022) | 80.00 | 1.44× | 14.7M | 49 | 10 |
| | **SNP-ZCA var (ours)** | **80.02** | 1.45× | 15.2M | **28** | 100 |
| **DeiT-Base** (81.84%) | VBP (Berisha et al., 2025) | 80.99 | 1.47× | 58.24M | – | – |
| | SAW (Berisha et al., 2025) | 78.88 | 1.47× | 58.24M | – | 0 |
| | SSP (Chen et al., 2021) | 80.08 | 1.62× | 56.8M | – | 100 |
| | VTP (Zhu et al., 2021) | 80.7 | 1.76× | 48.0M | – | 100 |
| | UVC (Yu et al., 2022) | 80.57 | 2.2× | – | – | 100 |
| | SAViT (Zheng et al., 2022) | **81.66** | 3.3× | 25.4M | 130 | 10 |
| | **SNP-ZCA var (ours)** | 81.58 | 3.3× | 23.2M | **75** | 100 |

Figure 6: **OPT (WikiText2)** The calibration set consists of 1024 sequences with sequence length 2048.

| Parameter reduction | OPT-125M | | OPT-1.3B | |
|---|---|---|---|---|
| | SliceGPT | Ours | SliceGPT | Ours |
| Uncompressed | 27.65 | | 14.62 | |
| 10% | 34.08 | **29.99** | 16.50 | **14.82** |
| 20% | 47.04 | **38.19** | 18.94 | **16.36** |
| 30% | 83.57 | **53.87** | 24.37 | **18.72** |
| 40% | 186.90 | **96.52** | 38.47 | **29.32** |
| 50% | **513.63** | 908.65 | **74.17** | 96.93 |

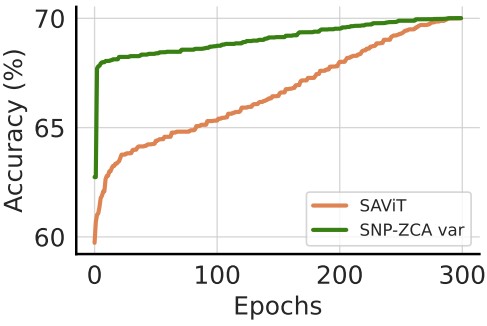

Figure 7: Performance recovery during retraining of pruned DeiT-Small networks.

Figure 7 shows another potentially useful property of our method. Due to our subspace reconstruction, our method regains performance significantly faster than SAViT. Concretely, the number of epochs required to reach 75% accuracy is almost 200 epochs greater for the SAViT method. We propose that this property may be leveraged to design retraining recipes that may reduce number of retraining epochs; however, we leave this exploration for future work.

Finally, we demonstrate our methods capability without retraining on OPT (Zhang et al., 2022b) language models. Table 6 shows the comparison of our SNP-ZCA var with SliceGPT (Ashkboos et al., 2024). We find that across pruning ratios and models, our method has a substantial advantage over the baseline. However, our performance seems to drop faster than that of SliceGPT when significant (50%+) of parameters are pruned.

## 5 DISCUSSION

In this work, we introduce a novel view of reconstructing neural activity during pruning. Research into such approaches holds promise for the potential future of pruning with minimal retraining and

without extensive search for the correct layer-wise pruning ratios. Further, we have shown that our method, derived from the subspace reconstruction, is consistently competitive or outperforms the state-of-the-art while being interpretable and computationally significantly less expensive.

However, a number of additional areas of exploration remain open. First, we find our proposed method to be somewhat successful when pruning networks without retraining. However, to reach competitive performance, retraining is nonetheless essential when significantly pruning networks. Future work should consider how to best make use of reconstructed network activity to ensure retraining is not necessary.

Second, a global variance-based cutoff to determine pruning ratios for all layers of a network. While this is a significant improvement, in terms of reducing manual tuning, over competitive methods, this method does not consider the downstream sensitivity of a network to pruning of upstream layers, implicitly assuming that all unit variances are equally important. Alternative importance measures which also focus on the differential importance of non-redundant neural contributions (within our proposed subspace) could improve this method further.

Finally, in this work, we restricted ourselves to examining the pruning of pre-trained networks. A promising extension of this work would be to prune networks during training and retraining. The most competitive methods that we could identify all make use of such a *during* (re)training adjustment of pruned units which allows for nodes to be gradually pruned rather than pruned all at once.

In conclusion, we see the subspace node pruning method described herein as a new, general, and effective perspective on the problem of pruning. This method can be combined with any existing node importance scoring method and has the potential to significantly improve the efficiency of node-pruning. Ultimately, this provides a new route forward for simple and efficient model speedup.

## REPRODUCIBILITY STATEMENT

In order to ensure reproducibility, we add the pseudocode of the algorithm in Algorithm 1. Furthermore, the Appendix E describes the models, datasets, and hyperparameters required for running our models. Lastly, we make available all code used to generate the experiments at <SEE ATTACHED ZIP>.

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

## A    RELATION TO LINEAR LEAST SQUARES

So far, we have introduced a method where pruning latent variables within a GS subspace corresponds to pruning unit inputs. This approach is based on the assumption that pruning within the subspace has minimal impact on the subsequent layer, without any notion of what 'minimal impact' means. In the following, we seek to demonstrate the efficacy of this subspace pruning by proving its equivalence to

LLS approximation for recovering the original inputs from their pruned counterparts. Thereby, we show that 'minimal impact' means to minimize the sum-squared difference of the unit inputs from their pruned counterparts. For clarity, we omit the layer subscript in the following, as the computation is fully contained within each layer.

We start by defining the recovery matrix $\mathbf{A}$ as

$$\arg\min_{\mathbf{A}}\|\mathbf{X} - \mathbf{A}\mathbf{X}_{(*,)}\|_2^2.$$

SNP uses a linear projection $\mathbf{M}$ to project inputs onto an orthogonal subspace. We may rewrite

$$\arg\min_{\mathbf{A}}\|\mathbf{X} - \mathbf{A}\mathbf{X}_{(*,)}\|_2^2 = \arg\min_{\mathbf{A}}\|\mathbf{X} - \mathbf{A}(\mathbf{M}_{(*,*)})^{-1}\hat{\mathbf{X}}_{(*,)}\|_2^2.$$

Solving for $\mathbf{A}(\mathbf{M}_{(*,*)})^{-1}$ by traditional LLS, we get

$$\mathbf{A}(\mathbf{M}_{(*,*)})^{-1} = \mathbf{X}(\hat{\mathbf{X}}_{(*,)})^\top(\hat{\mathbf{X}}_{(*,)}(\hat{\mathbf{X}}_{(*,)})^\top)^{-1}.$$

We observe that our latent variables $\hat{\mathbf{X}}$ are orthogonal by definition of SNP. Therefore, we may rewrite the equation using $\hat{\mathbf{X}}_{(*,)}(\hat{\mathbf{X}}_{(*,)})^\top = \mathbf{D}_{(*,*)}$. Note that pruning the row dimension of the left and column dimension of the right matrix in a product may be expressed by pruning its product in both dimensions.

$$\mathbf{A}(\mathbf{M}_{(*,*)})^{-1} = \mathbf{X}(\hat{\mathbf{X}}_{(*,)})^\top(\mathbf{D}_{(*,*)})^{-1} = \mathbf{M}^{-1}\hat{\mathbf{X}}(\hat{\mathbf{X}}_{(*,)})^\top(\mathbf{D}_{(*,*)})^{-1},$$

where we used the definition of our GS transformation to obtain the last equation. Given that $\hat{\mathbf{X}}(\hat{\mathbf{X}}_{(*,)})^\top = \mathbf{D}_{(,*)}$,

$$\mathbf{A}(\mathbf{M}_{(*,*)})^{-1} = \mathbf{M}^{-1}\mathbf{D}_{(,*)}(\mathbf{D}_{(*,*)})^{-1} = \mathbf{M}^{-1}\mathbf{I}_{(,*)} = (\mathbf{M}^{-1})_{(,*)},$$

where $\mathbf{I}$ is the identity matrix. We find that a matrix $\mathbf{A} = (\mathbf{M}^{-1})_{(,*)}\mathbf{M}_{(*,)}$ optimally minimizes the squared approximation error of the unpruned inputs and the approximation from the pruned inputs. Notably, the recovery matrix $\mathbf{A}$ is precisely equivalent to the product of the pruned subspace transformation matrices as outlined earlier. Consequently, instead of computing the LLS approximation, the same recovery matrix can be obtained by employing the SNP method.

With this equivalence established, we demonstrate that our method optimally approximates the original inputs from their pruned counterparts in a linear manner, thus proving the efficacy of our approach in reducing the error induced by pruning. Moreover, it underscores that the approximation process is independent of the specific ordering of units within the pruned and retained sets – a result that is not immediately apparent from our approach of pruning within a ranked subspace.

Compared to the existing literature, our method focuses on recovering the inputs, whereas approaches by Mariet & Sra (2015) and He et al. (2017) employ LLS to derive a new weight tensor. Despite this distinction, the resulting reparameterized weights are fundamentally equivalent. If we recast the LLS problem in their framework as one of approximating the layer outputs from pruned inputs, we seek to optimize a novel weight matrix $\hat{\mathbf{W}}$ through the following minimization:

$$\arg\min_{\hat{\mathbf{W}}}\|\mathbf{Y} - \hat{\mathbf{W}}\mathbf{X}_{(*,)}\|_2^2.$$

By decomposing $\hat{\mathbf{W}} = \mathbf{W}\mathbf{A}$, we may equivalently optimize for $\mathbf{A}$ in:

$$\arg\min_{\mathbf{A}}\|\mathbf{W}\mathbf{X} - \mathbf{W}\mathbf{A}\mathbf{X}_{(*,)}\|_2^2.$$

The remainder of the proof follows trivially from our proof above, with the only difference being the inclusion of the weight term. The resulting recovery matrix $\mathbf{A}$ is identical to that obtained before, thereby demonstrating that these methods for activity recovery are equivalent.

By revealing these insights, we further solidify the robustness and generality of our method.

## B  ALGORITHM INCLUDING PERMUTATIONS

In Section 2.1, we proposed to prune units in an orthogonal subspace within which pruning minimally impacts the layers' activity. In Algorithm 1, we detail the computational steps to run the method.

Section 2.2 extends this method by including a permutation matrix that re-organizes the data prior to subspace node pruning. Note that the permutation matrix is defined based upon any additional importance scoring which is combined with our method.

We adapt Algorithm 1 to choose a particular importance score, by simply permuting the input features in $\mathbf{X}_l$, such that the features are sorted from most- to least important. In order to keep the adjacent layers unaffected, we unpermute the subspace transformation and its inverse after pruning and prior to the weight matrix multiplication. See Algorithm 2 for the algorithm description.

---

**Algorithm 2:** Layer-wise subspace node pruning with permutation

---

**Input:** Data $\mathbf{X}_l$, Permutation Matrix $\mathbf{P}_l$, Weights $\mathbf{W}_l$, Number of units to prune $n$
**Output:** Pruned weights $\hat{\mathbf{W}}_l$
$\mathbf{C}_l = (\mathbf{P}_l\mathbf{X}_l)(\mathbf{P}_l\mathbf{X}_l)^T$ ▷ Compute dot-product between (permuted) input feature vectors
$\mathbf{M}_l^{-1}, \mathbf{D}_l = \text{LDL}(\mathbf{C}_l)$ ▷ Decompose matrix $\mathbf{C}$
$\hat{\mathbf{W}}_l = \mathbf{W}_l\mathbf{P}_l^{-1}\mathbf{M}_{l,(:,:n)}^{-1}\mathbf{M}_{l,(:n,:n)}\mathbf{P}_{l(:n,*)}$ ▷ Prune $\mathbf{M}$ and $\mathbf{M}^{-1}$ (leading to pruned $\mathbf{W}_1$)
▷ Note: * indicates non-zero columns only

**Return:** $\hat{\mathbf{W}}_l$

---

## C  LOCAL RECONSTRUCTION-LOSS

Given a matrix $\mathbf{T} \in \mathcal{R}^{n \times m}$, with $m \geq n$, we may ask ourselves, which column is most redundant relative to all other columns? Note that here we assume a metric for 'redundancy', specifically the Frobenius norm of the residual of a best linear fit of each column using all others. In other words, which column is best linearly approximated by the others?

We can formulate this as the optimization problem

$$\delta t^* = \underset{\delta t}{\text{argmin}}\frac{1}{2}\|\mathbf{T}e_k - \mathbf{T}\delta t\|_2^2 \qquad \text{s.t. } e_k^\top \delta t = 0,$$

where $e_k$ is a unit-vector with a one in the $k^{\text{th}}$ position. The condition $e_k^\top \delta t = 0$ then ensures that a solution does not use the $k^{\text{th}}$ column for reconstruction.

To solve this constrained optimization problem, we define a Lagrange multiplier, such that

$$\mathcal{L}_k = \frac{1}{2}\|\mathbf{T}e_k - \mathbf{T}\delta t\|_2^2 + \lambda e_k^\top \delta t.$$

We find the minimizer using the first derivative test of this convex loss.

$$\frac{\partial \mathcal{L}_k}{\partial \delta t} = -\mathbf{T}\mathbf{T}^\top e_k + \mathbf{T}\mathbf{T}^\top \delta t + \lambda e_k$$
$$\implies \delta t^* = e_k - \lambda[\mathbf{T}\mathbf{T}^\top]^{-1}e_k$$

Now, we do the same wrt. $\lambda$ and use our solution for $\delta t$

$$0 = e_k^\top \delta t^* = e_k^\top \left(e_k - \lambda[\mathbf{T}\mathbf{T}^\top]^{-1}e_k\right)$$
$$= 1 - \lambda e_k^\top[\mathbf{T}\mathbf{T}^\top]^{-1}e_k$$
$$\implies \lambda = \frac{1}{e_k^\top[\mathbf{T}\mathbf{T}^\top]^{-1}e_k}$$

Finally, we evaluate the reconstruction loss of our unit $k$:

$$\mathcal{L}_k^* = \frac{1}{2}\|\mathbf{X}e_k - \mathbf{T}\delta t^*\|_2^2$$

$$= \frac{1}{2}\|\mathbf{T}e_k - \mathbf{T}\left(e_k - \lambda[\mathbf{T}\mathbf{T}^\top]^{-1}e_k\right)\|_2^2$$

$$= \frac{1}{2}\| - \lambda[\mathbf{T}\mathbf{T}^\top]^{-1}e_k\|_2^2$$

$$= \frac{1}{2}\|\frac{\mathbf{T}[\mathbf{T}\mathbf{T}^\top]^{-1}e_k}{e_k^\top[\mathbf{T}\mathbf{T}^\top]^{-1}e_k}\|_2^2$$

$$= \frac{1}{2}\frac{e_k^\top[\mathbf{T}\mathbf{T}^\top]^{-1}\mathbf{T}\mathbf{T}^\top[\mathbf{T}\mathbf{T}^\top]^{-1}e_k}{\left(e_k^\top[\mathbf{T}\mathbf{T}^\top]^{-1}e_k\right)^2}$$

$$= \frac{1}{2}\frac{e_k^\top[\mathbf{T}\mathbf{T}^\top]^{-1}e_k}{\left(e_k^\top[\mathbf{T}\mathbf{T}^\top]^{-1}e_k\right)^2}$$

$$= \frac{1}{2}\frac{1}{e_k^\top[\mathbf{T}\mathbf{T}^\top]^{-1}e_k}$$

$$= \frac{1}{2}\frac{1}{([\mathbf{T}\mathbf{T}^\top]^{-1})_{kk}}$$

Now, if we choose $\mathbf{T} = \mathbf{R}\mathbf{X}$, we find our unnormalized-ZCA metric

$$\mathcal{L}_k = \frac{1}{2}\frac{1}{([\mathbf{R}\mathbf{X}\mathbf{X}^\top\mathbf{R}^\top]^{-1})_{kk}} = \frac{1}{2}S_{kk}^2,$$

where we choose units based on the lost variance within the subspace.

Going one step further, we can choose $\mathbf{T} = \mathbf{X}$ resulting in

$$\mathcal{L}_k = \frac{1}{2}\frac{1}{([\mathbf{X}\mathbf{X}^\top]^{-1})_{kk}}.$$

This chooses units based on the units whose variances outside the subspace, but after reconstruction by all remaining units impacted the layer's input minimally.

Frantar & Alistarh (2023) are going even further by taking weight transformation into account. Therefore, they estimate importance based on the change in the layer's output by pruning individual parameters.

Lastly, Molchanov et al. (2016; 2019) expand the loss of networks based on first- and second-order Taylor expansions and compute the sensitivity of the loss to the removal of individual units. Similarly, Wang et al. (2019) prunes based on a second-order Taylor expansion of the loss, while approximating the Hessian term by the Kronecker factorization Martens & Grosse (2015). These approaches are assumed to produce globally calibrated importance scores due to the approximation of the loss change. However, the issue is that their importance measures overestimate the ability of the network to reconstruct each unit when pruning in a single shot. In contrast, our global calibration properly recomputes the local importances.

## D    RECONSTRUCTION-AWARE IMPORTANCE SCORING

In the main text, we have discussed a reordering strategy based on the diagonal elements of the un-normalized ZCA matrix that describe the latent variances in an orthogonal subspace. The orthogonal subspace ensures that we only estimate importance from information that we cannot recover via LLS and our reconstruction method. Here, we show that, instead of a ranking based on the variances, we can use a wider range of importance scoring methods while disregarding recoverable information. In particular, we demonstrate how the simple summed absolute weights (SAW) importance measure possibly benefits from this idea.

First, we need to note that we cannot reconstruct any activity from unit inputs that are already orthogonal. If all inputs were orthogonal, $\mathbf{X}_l = \hat{\mathbf{X}}_l$ and subsequently $\mathbf{M}_l = \mathbf{I}_l$. Therefore, if we assumed all inputs to be orthogonal, we could prune with minimal impact on the subsequent layer and consequently estimate importance without considering the unit activity that we may recover.

In the following, we assume that any weight tensor may be decomposed into an input orthonormalization matrix and a transformation of the latent variables within that orthonormal subspace. We have $\mathbf{W}_l = \tilde{\mathbf{W}}_l (\mathbf{X}_l \mathbf{X}_l^\top)^{-\frac{1}{2}} = \tilde{\mathbf{W}}_l \boldsymbol{\Sigma}_l^{-\frac{1}{2}}$, with the orthonormalization matrix $\boldsymbol{\Sigma}_l^{-\frac{1}{2}}$ obtained by ZCA and the underlying weight transformation to the latent variables $\tilde{\mathbf{W}}_l$ (Ahmad, 2024). Therefore, given a linear layer parameterized as $\mathbf{Y}_{l+1} = \mathbf{W}_l \mathbf{X}_l$, we equivalently write as $\mathbf{Y}_{l+1} = \tilde{\mathbf{W}}_l \boldsymbol{\Sigma}_l^{-\frac{1}{2}} \mathbf{X}_l$. A look at the cross-correlations of the unit outputs

$$\mathbf{Y}_{l+1} \mathbf{Y}_{l+1}^\top = \tilde{\mathbf{W}}_l \boldsymbol{\Sigma}_l^{-\frac{1}{2}} \mathbf{X}_l \mathbf{X}_l^\top (\boldsymbol{\Sigma}_l^{-\frac{1}{2}})^\top \tilde{\mathbf{W}}_l^\top = \tilde{\mathbf{W}}_l \tilde{\mathbf{W}}_l^\top \,,$$

reveals that they are now fully defined by the transformation $\tilde{\mathbf{W}}_l$. Hence, our proposal to measure the SAW measure of $\tilde{\mathbf{W}}_l = \mathbf{W} \boldsymbol{\Sigma}_l^{\frac{1}{2}}$ instead of the weights.

Now, we compare the efficacy by pruning VGG networks without retraining. We refer to this novel method as SNP-SAW-tilde and compare its performance to the original SNP-SAW before retraining.

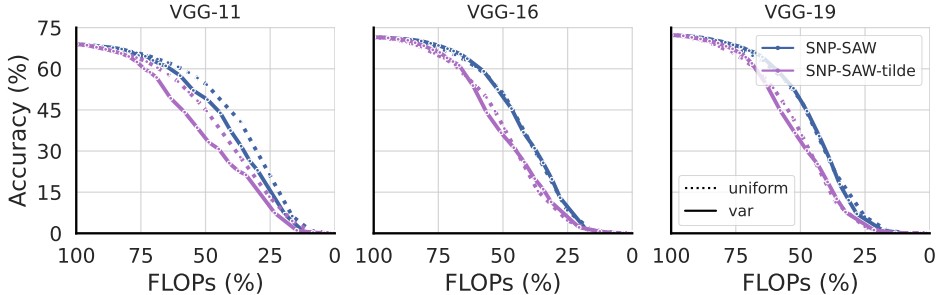

Figure 8: Similar to Figure 4, we compare SNP-SAW-tilde against SNP-SAW on VGG-16 without retraining.

Figure 8 shows that this novel method of importance scoring cannot keep up with the good performance of SNP-SAW.

Since this measure is still a very novel measure, it requires further analysis, including retraining. Compared to SNP-SAW, our novel scoring first removes the scaling inherent to each unit, such that all latent variables have unit scaling. Therefore, the scaling measured by SAW is independent of the initial scaling of the inputs. This is different from SNP-SAW and SNP-ZCA which both perform superior to all our other methods after retraining. We hypothesize that a measure of this unit scaling may increase the performance. Furthermore, ZCA is a way of all-to-all orthonormalization. Therefore, this overestimates ability to reconstruct upon pruning as pruned units no longer contribute to the orthonormalization. This overestimation may be, albeit very costly, circumvented by recomputing the measure after each pruning step. Note that this equally applies to our SNP-ZCA measure.

While this measure has not yet been proven to improve upon other measures evaluated herein, it is certainly interesting for future work.

## E  EXPERIMENTAL DETAILS

**Models**  For our demonstrations on the VGG networks and ResNet-50, we prune the pre-trained networks from PyTorch (Paszke et al., 2019) on the ILSVRC (ImageNet) dataset. Pre-trained DeiT models are loaded using the Timm library (Wightman, 2019), and the OPT models from Hugging Face (Wolf et al., 2019).

**VGG-16 baselines**  Several comparison methods are re-implemented, including summed absolute weights (SAW) (Li et al., 2016), ThiNet (Luo et al., 2017) and PFA-EN (Cuadros et al., 2020). SAW

and ThiNet assume that a practitioner might uniformly prune all layers by the same amount. This global pruning style is referred to as an 'uniform' pruning ratio. Both SAW and ThiNet provide no guidance on the global ranking of units, instead assuming that a practitioner might uniformly prune all layers by the same amount. Therefore, we employ a uniform pruning ratio across layers when implementing these methods, referred to as a uniform pruning (uni) in all relevant figures. In contrast, PFA-EN performs PCA to decide on a global ranking of units on top of their local importance structure. Notably, we apply PFA-EN at the input nodes, rather than the output activations, finding that this produces the best performance. We re-implemented these baselines due to unavailable code or outdated packages and have code available for reproduction of all experiments at <SEE ATTACHED ZIP>.

**Retraining recipe** The retraining recipe is a significant contributing factor to network performance after retraining (Wang et al., 2023). In order to keep fair comparisons, we re-implemented a few important baselines in the literature on VGG-16 and compared their performance under the same retraining recipe. For VGG-16 we empirically found that our models performed best under the initial training recipe for the networks by (Paszke et al., 2019).

For ResNet-50, we used a modified recipe of Wang et al. (2023) and Wang & Fu (2023) to ensure a comparison to the novel methods under the same retraining recipe. Table 3 shows these retraining recipes that we used. For measuring the Gram matrix of every layer's activations (inputs), we make use of the full training set images transformed via the test-transforms. Note that for computing layer-wise cross-correlations of the inputs, as well as for validating model performance, we used the test-transformations. I.e. no data augmentation beyond resizing and center-cropping the images, as well as normalizing by the mean and standard deviation shown in the table. These transforms are the default test-transforms that were used to evaluate the pre-trained models.

Table 3: Hyperparameter overview for retraining VGG-16, ResNet-50 on ImageNet, and ResNet-56 on CIFAR-10.

| Hyperparameter | VGG-16 | ResNet-50 | ResNet-56 (CIFAR-10) |
|---|---|---|---|
| Steps at Epochs | 30 / 60 | 30 / 60 / 75 | 60 / 80 |
| Optimizer | SGD with momentum | | |
| Learning Rate | 0.01 | | 0.01 |
| Momentum | 0.9 | | |
| Weight Decay (L2 Regularization) | $1 \times 10^{-4}$ | | $5 \times 10^{-4}$ |
| Batch Size | 256 | | 128 |
| Number of Epochs | 90 | | 100 |
| Learning Rate Scheduler | MultiStepLR | | |
| Learning Rate Decay Factor | 0.1 | | |
| Data Augmentation | RandomResizedCrop RandomHorizontalFlip | | RandomCrop RandomHorizontalFlip |
| Normalization | Mean: [0.485, 0.456, 0.406] Std: [0.229, 0.224, 0.225] | | Mean: [0.4914, 0.4822, 0.4465] Std: [0.2023, 0.1994, 0.2010] |
| Loss Function | CrossEntropyLoss | | |

For retraining DeiT models, we needed to slightly modify the retraining recipe developed for SAViT (Zheng et al., 2022). Since we retain a significant amount of accuracy due to our subspace reconstruction, large learning rates may destroy any performance retained. For DeiT-Tiny and -Small we empirically found learning rates of $1.25e^{-5}$ and $7.5e^{-5}$, respectively. Furthermore, we observed that a *momentum calibration*, that is setting up momentum terms without updating any model parameter during a first epoch, to be more effective than a slow warm-up. All other parameters are taken from SAViT.

**Pruning Ratios** For our analysis on VGG networks without retraining, we incremented the global pruning ratios in steps of 0.01 until 0.1, 0.025 until 0.3 and then used steps of 0.1. For ThiNet, we used a step size of 0.1 throughout. For our retraining analysis, we used the following pruning ratios for uniform pruning: [0.1, 0.2, 0.3, 0.4, 0.5]. Table 4 shows the pruning ratios for the variance and PCA heuristics. We used the ratio that was closest in terms of FLOP count to the uniform pruning ratios.

Similarly, for multi-branch networks, we determined the desired pruning ratio by increments of 0.01 and choosing the ratio that was closest to the desired FLOP speedup ratios.

Table 4: Pruning ratios for VGG-16 retraining experiments.

| Method | Pruning ratios |
|--------|----------------|
| SAW | [0.04, 0.1, 0.175, 0.25, 0.3] |
| ZCA | [0.03, 0.8, 0.125, 0.2, 0.275] |
| PFA-EN | [0.01, 0.04, 0.06, 0.1, 0.15] |

## F  RESNET AND TRANSFORMER PRUNING

To prune networks with skip connections, we adapt the Dependency Graph (DepGraph) framework (Fang et al., 2023). DepGraph groups layers so that when a node is pruned in one layer, the corresponding nodes in related layers are pruned as well. In ResNets, information flows through two parallel branches: the residual branch and the main branch. These branches merge by summing their outputs element-wise, such that both pathways must have the same output dimensionality to fully take advantage of pruning a node. DepGraph ensures that when a node is pruned, the corresponding input and output connections are also removed in computationally adjacent layers. Consequently, it is not clear which layer of a group to use for importance scoring. In this work, we prune nodes based upon the input activity. Naturally, this extends to scoring group-wise input activities for our unnormalized-ZCA importance and the global variance-based pruning cutoffs.

Next to residual connections, DeiT networks have a number of distinct attention heads. We aggregate the importances of the heads and take the average, such that each head is pruned with the same pruning ratio. This allows for efficient parallel computation, without removing an entire head. Last but not least, we do not prune the patch embedding layer.

## G  PRUNING VGG NETWORKS WITHOUT RETRAINING

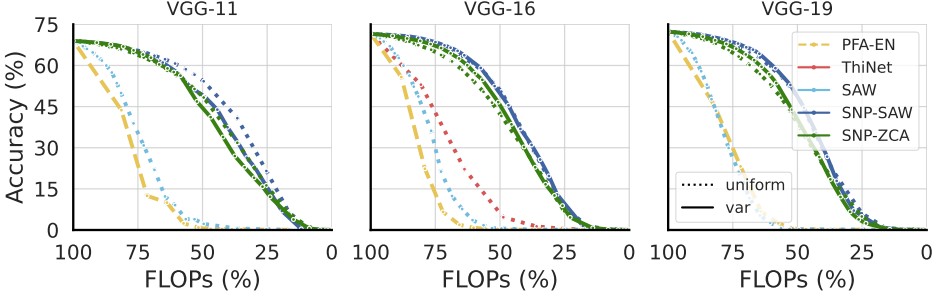

Figure 9: Pruning performance of VGG-11/16/19 without retraining. Test accuracy against reduction in FLOPs. We compare baseline pruning methods to both our proposed method (SNP-ZCA) and to the combination of our subspace-construction method with the SAW importance ordering (SNP-SAW). Furthermore, we show results both for uniform pruning per-layer (dashed lines) and our proposed global variance-based pruning (solid lines).

## H    VGG-16 TABLE

Table 5: **VGG-16 (ImageNet)**: The final accuracies, FLOP count, and Parameter count for our method along with re-implemented alternative methods applied to the VGG-16 network. See Figure 4 for these results in training curve form. To ensure a fair comparison of our results and those in the literature, we group the results into blocks with similar FLOP speed-ups. Each of these blocks naturally has results which are obtained by pruning at different ratios. Groups are separated by dashed lines.

| Importance | Final %Acc | ΔAcc | FLOP speedup | #Params |
|---|---|---|---|---|
| ThiNet (reimpl.) | 71.36 | -0.23 | 2.02× | 68.79M |
| PFA-EN (reimpl.) | 71.69 | 0.10 | 1.97× | 76.97M |
| SAW (reimpl.) | 71.58 | -0.01 | 2.02× | 68.79M |
| **SNP-SAW uniform** (ours) | 71.50 | -0.09 | 2.02× | 68.79M |
| **ZCA var** (ours) | 72.03 | 0.44 | 1.97× | 83.89M |
| **SNP-SAW var** (ours) | 71.67 | 0.08 | 2.14× | 75.58M |
| **SNP-ZCA var** (ours) | **72.08** (±0.02) | **0.49** | 1.97× | 83.89M |
| ThiNet (reimpl.) | 69.68 | -1.91 | 2.74× | 50.93M |
| PFA-EN (reimpl.) | 70.11 | -1.48 | 2.75× | 53.58M |
| SAW (reimpl.) | 70.02 | -1.57 | 2.74× | 50.93M |
| **SNP-SAW uniform** (ours) | 70.00 | -1.59 | 2.74× | **50.93M** |
| **ZCA var** (ours) | 70.71 | -0.88 | 2.75× | 63.05M |
| **SNP-SAW var** (ours) | 70.53 | -1.06 | 2.90× | 57.57M |
| **SNP-ZCA var** (ours) | **70.77** (±0.06) | **-0.82** | 2.75× | 63.05M |

## I    DEIT PRUNING WITH SUBSAMPLING

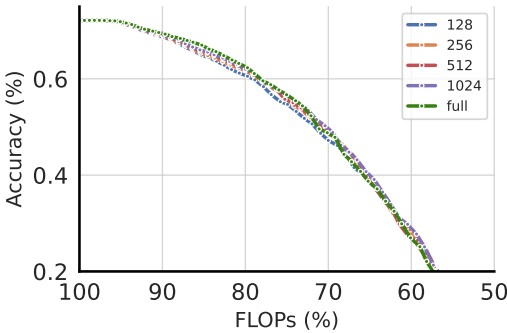

Figure 10: DeiT-tiny accuracy after pruning. Each line shows the pruning performance when the Gram matrix is computed from a different number of samples randomly drawn from the training set. 'full' means using the entrie imagenet training set.

## J    RESNET-56 ON CIFAR-10 PRUNING

Table 6: **ResNet-56 (CIFAR-10)**: We report the mean and standard deviations over three retraining runs with different seeds. The columns represent the FLOP reduction over the original network.

| | 1.5× | 2.0× | 2.5× | 3.0× |
|---|---|---|---|---|
| SAW | 93.72 ± 0.15 | 93.31 ± 0.14 | 92.97 ± 0.09 | 92.64 ± 0.01 |
| ZCA var | 93.66 ± 0.10 | **93.49** ± 0.05 | 93.03 ± 0.04 | 92.75 ± 0.11 |
| SNP-ZCA var | **93.81** ± 0.06 | 93.40 ± 0.10 | **93.08** ± 0.13 | **92.97** ± 0.08 |

# K   COMPARISON TO LOW-RANK APPROXIMATION

Due to the similarity of our proposed method of pruning within a GS subspace to pruning in any other linear subspace, such as one produced via PCA, we compare the approaches in this section.

In particular, we compute the low rank approximation by computing the eigen-decomposition of our Gram matrices $\mathbf{XX}^{\top} = \mathbf{U}\mathbf{\Lambda}\mathbf{U}^{T}$. We use the eigenvectors $\mathbf{U}$ to map our activations into the subspace, choose the variance cutoff for pruning within this subspace based on the eigenvalues, $\mathrm{diag}(\mathbf{\Lambda})$, and transform back to the original space via the inverse of the eigenvectors.

Our reconstruction matrix is given by $\mathbf{A}_{\text{low rank}} = (\mathbf{U}^{\top})_{(,*)}\mathbf{U}_{*}$, and can be combined with the weights as

$$\hat{\mathbf{W}} = \mathbf{W}(\mathbf{U}^{\top})_{(,*)}\mathbf{U}_{*}.$$

Unfortunately, the eigenvectors have no lower-triangular structure, such that the reconstruction matrix is not pruned in its shape, but merely in its rank. This easily results in the sub-space pruning counterintuitively causing an increase in parameters. In order to generate a real-world speed-up at a certain level of sparsity, we re-parameterize each low-rank layer as two smaller matrices $\mathbf{W}^{(1)} = \mathbf{U}_{*}$ and $\mathbf{W}^{(2)} = \mathbf{W}(\mathbf{U}^{\top})_{(,*)}$.

Figures 11 and 12 on the left show that low-rank approximation maintains accuracy to a greater degree and has a large advantage over our proposed SNP method when considering the number of intrinsic dimensions. However, as the other two panels show, this advantage does not translate into real world gains either in inference latency (wall-clock time) or parameter count. This demonstrates the practical significant of node-pruning methods over rank-reducing methods.

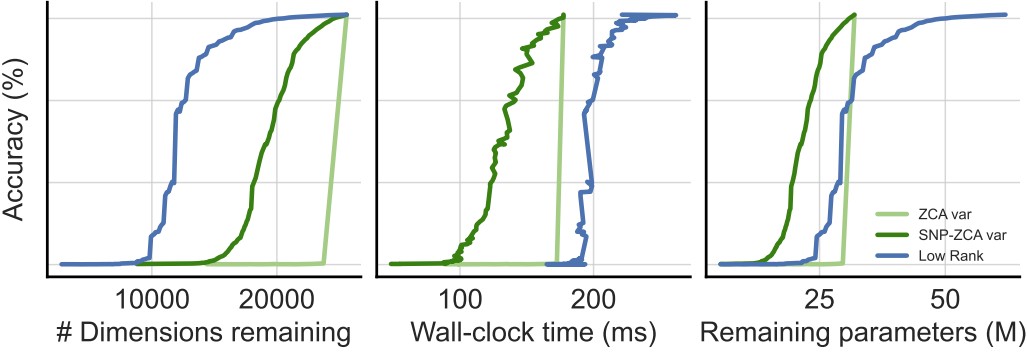

Figure 11: **ResNet-50 (ImageNet)**: We compare our proposed method of pruning within a GS subspace (SNP-ZCA var) to pruning within a PCA subspace (Low Rank) which leads to low-rank approximation, rather than pruning. We further added a comparison to our importance scoring without reconstruction (ZCA var). We report the performance without retraining. The number of dimensions remaining refer to the number of units remaining for our method, whereas in low rank approximation, it refers to the number of intrinsic dimensions remaining. Wall-clock time is measured over 5 trials on an A100 GPU with batch size 512. We report mean and standard deviation, although the latter are so small, they are barely visible.

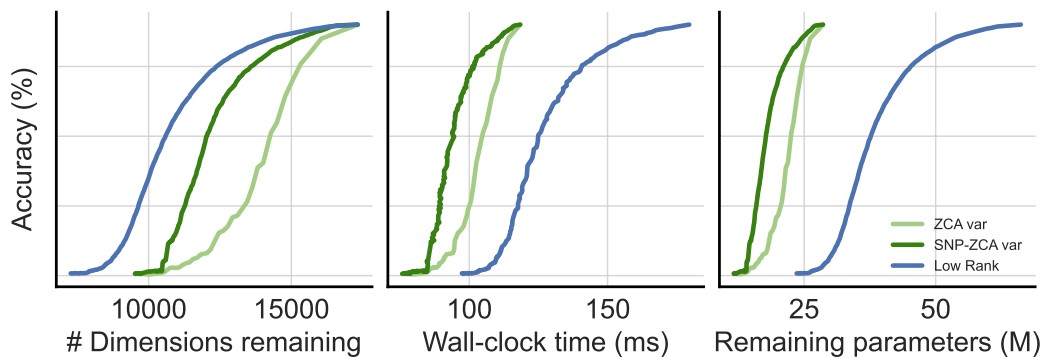

Figure 12: **DeiT-Tiny (ImageNet)**: Same comparisons as Figure 11 but with a Vision Transformer Model.

## L    RELATION OF FLOPS AND WALL-CLOCK TIME

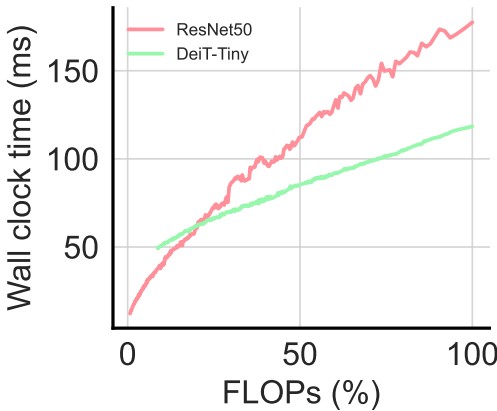

Figure 13: We relate the wall-clock time of our method (SNP-ZCA var) against the remaining number of FLOPs of the network. We report the mean and standard deviation over 5 trials on an A100 GPU with batch size 512. Standard deviations are extremely minimal and therefore barely visible.

