# OpenReview forum: "Subspace Node Pruning: Revealing Structured Importance via Orthogonal Subspace Transformations"
_ICLR.cc/2026/Conference — Submitted to ICLR 2026_

### Official Review · Reviewer_ARux · 2025-10-29

**Soundness:** 2
**Presentation:** 2
**Contribution:** 2
**Rating:** 4
**Confidence:** 3

**Summary:**

The paper proposes Subspace Node Pruning (SNP), a structured pruning framework that projects layer inputs into a lower-triangular orthogonal subspace (via LDL/Gram-Schmidt), prunes the least informative latent units, and automatically reconstructs the dropped contributions through a linear least-squares–equivalent reparameterization of the weights. It complements this with (i) an unnormalized-ZCA–based importance score that ranks units by their non-redundant (post-orthogonalization) variance, and (ii) a global cumulative-variance cutoff that automatically allocates layer-wise pruning ratios.

**Strengths:**

(1) The paper is well-motivated. The core idea of orthogonalization-based node pruning comes with a clear, coherent technical narrative.

(2) Global calibration of pruning across layers via a single cutoff/ordering is an attractive and practical design choice.

(3) Empirical results are competitive on standard CNNs (VGG, ResNet); the pre-ordering idea shows benefits in the reported settings.

**Weaknesses:**

(1) Experimental breadth is still limited for a pruning paper: more diverse architectures (e.g., mobile/dense variants) and tasks/datasets (e.g., CIFAR-10/100, generative models) would improve generality; reporting actual inference-time/throughput speed ups on unified hardware is also important.

(2) Missing or incomplete baselines: SVD/low-rank pruning is a natural comparison given the conceptual proximity; related modern structured-pruning baselines could be covered more exhaustively.

(3) Improvements over strong baselines are often marginal, and in some ResNet settings the method is not SOTA. Please contextualize gains (effect sizes, significance) and discuss trade-offs.

(4) Presentation issues: some formatting/typos and unclear definitions detract from readability. For example, “howver” in line 367.

**Questions:**

(1) Can you report end-to-end inference latency and throughput (same hardware, batch size, backend) and relate them to FLOPs reductions? This would substantiate practical speedups.

(2) Please clarify grouped/joint importance computation for ResNets (exact procedure/algorithmic steps) and discuss computational cost when moving from simple feed-forward layers to CNN/ResNet blocks.

---

> ### Author Response · Authors · 2025-11-21
>
> We thank the reviewer for their feedback and questions.
>
>
> ```Experimental breadth is still limited for a pruning paper: more diverse architectures (e.g., mobile/dense variants) and tasks/datasets (e.g., CIFAR-10/100, generative models) would improve generality; reporting actual inference-time/throughput speed ups on unified hardware is also important.```
>
> We aimed to cover experimental breadth by considering traditional CNNs (VGG), ResNets, Vision Transformers, and Language Transformer Models. We further ran our method on CIFAR-10 with the results shown below (also added as Appendix J in our revised manuscript).
>
> ResNet-56 on CIFAR-10:
>
> -|1.5 $\times$|2.0 $\times$|2.5 $\times$|3.0 $\times$
> --|--|--|--|--
> SAW|$93.72 \pm 0.15$|$93.31 \pm 0.14$|$\underline{92.97} \pm 0.09$|$92.64 \pm 0.01$
> ZCA var|$93.66 \pm 0.10$|$\mathbf{93.49} \pm 0.05$|$\underline{93.03} \pm 0.04$|$92.75 \pm 0.11$
> SNP-ZCA var|$\mathbf{93.81} \pm 0.06$|$93.40 \pm 0.10$|$\mathbf{93.08} \pm 0.13$|$\mathbf{92.97} \pm 0.08$
>
>
> ```Missing or incomplete baselines: SVD/low-rank pruning is a natural comparison given the conceptual proximity; related modern structured-pruning baselines could be covered more exhaustively.```
>
> SVD based or low-rank pruning can be seen as a natural comparison due to conceptual similarity between these methods and our subspace construction.
> Notably, however, pruning in a low-rank fashion often does not translate into true computational speed (FLOP) improvements.
> To make clear the tradeoffs of using our subspace node pruning method versus a low-rank approach, we have added Appendix K to the revised manuscript.
> Appendix K demonstrates that low-rank (via SVD) methods do very well in maintaining accuracy through pruning when viewed in terms of the subspace-dimensions which remain.
> However, the transformations to and from the subspace are dense, rather than lower-triangular as they are for SNP.
> This means that low-rank pruning is practically much worse than our proposed SNP method when the wall-clock computational time, or actual remaining parameter number are considered.
>
> ```Improvements over strong baselines are often marginal, and in some ResNet settings the method is not SOTA. Please contextualize gains (effect sizes, significance) and discuss trade-offs.```
>
> Note, that the power of this method is particularly in it's simplicity of application and lack of hyperparameter tuning requirement for the compression ratio of every network layer.
> Given this, the ability for it to match SOTA is extremely impressive by our estimation.
> We refer the reviewer to our response to reviewer Rnnj for a more detailed explanation.
>
> ```Presentation issues: some formatting/typos and unclear definitions detract from readability. For example, “howver” in line 367.```
>
> Thank you for this note, we have thoroughly proof edited the manuscript for the new revision.
>
>
> ```Can you report end-to-end inference latency and throughput (same hardware, batch size, backend) and relate them to FLOPs reductions? This would substantiate practical speedups.```
>
> To demonstrate practical speed ups we added Appendix L to our revised manuscript. Here, wall-clock time (inference latency for a single batch) is shown on an A100 GPU for our method applied to two models, ResNet50 and DeiT-Tiny.
> Importantly, they both show that our calculation of FLOP improvement has a linear relationship with inference latency and truly translates to practical speedups.
>
> ```Please clarify grouped/joint importance computation for ResNets (exact procedure/algorithmic steps) and discuss computational cost when moving from simple feed-forward layers to CNN/ResNet blocks.```
>
> We use the Dependency Graph[1] for creating pruning groups in ResNets.
> That is, we group all layers whose computational requirement is that they should have the same input or output shape.
> In each group, we use the first layer that is pruned in its input dimension, i.e. the layer pruned in its input dimension that is closest to the input. This is a purely empirical choice as we have observed little difference in the performance of selecting these based on the averaged group importance or the last layer.
> Therefore, the computational cost for moving beyond single branch models is limited to a single forward pass the Dependency Graph package needs for analyzing the network structure, as well as importance scoring based upon selected layers in the graph.
>
> [1] Fang, G., Ma, X., Song, M., Mi, M. B., & Wang, X. (2023). Depgraph: Towards any structural pruning. In Proceedings of the IEEE/CVF conference on computer vision and pattern recognition (pp. 16091-16101).

---

### Official Review · Reviewer_Rnnj · 2025-10-29

**Soundness:** 3
**Presentation:** 3
**Contribution:** 3
**Rating:** 6
**Confidence:** 3

**Summary:**

This paper introduces Subspace Node Pruning (SNP), a novel structured pruning method. The approach works by projecting neural network activations onto an orthogonal subspace using a lower-triangular transformation matrix. This unique transformation allows for pruning nodes in the subspace (which corresponds to pruning the original input nodes) while also automatically reconstructing lost, non-redundant information, via a proven equivalence to linear least squares (LLS).

The method includes two key complementary contributions:
- An "unnormalized-ZCA" scoring method to optimally reorder units based on their redundancy before pruning.
- A "cumulative variance" metric that acts as a global, automated mechanism for selecting non-uniform, layer-wise pruning ratios from a single, universal threshold.

Experiments on VGG-16, ResNet-50, DeiT, and OPT demonstrate that SNP achieves highly competitive performance against SOTA methods, while being significantly more computationally efficient (e.g., up to 24x faster than alternative pruning algorithms).

**Strengths:**

- The primary strength is the method's 24x speedup in the pruning step compared to SOTA iterative methods like TPP.
- The "cumulative variance" heuristic is an effective method for automatic, non-uniform, layer-wise ratio selection, without requires expert tuning.
- The LLS-equivalent reconstruction provides excellent pre-retraining accuracy (Fig 4 left, Fig 5), which means strong results on one-shot LLM pruning (Table 6).
- The method proposed in the paper has a wide range of application scenarios, from CNNs (VGG), ResNets, to Transformers (DeiT, OPT).

**Weaknesses:**

- The reconstruction provides a "rather small impact" after full retraining, could means that the LLS algorithm is less critical to the final result.
- The results are highly competitive but do not universally "match or exceed" SOTA. Its strength is the trade-off, not setting new accuracy records.
- The global variance heuristic is effective but limited, as it ignores downstream layer sensitivity, treating all variance as equally important.
- This method relies on computing and decomposing the Gram matrix $C_l = X_l X_l^T$, which has dimensions $n_l \times n_l$ (where $n_l$ is the number of units/filters), the $O(n_l^3)$ cost of decomposition (and $O(n_l^2)$ for storage) could become a bottleneck for layers with an extremely large number of units.
- Retraining is still necessary. Therefore, it is not a perfect one-shot solution.
- While the method automates layer-wise ratios, it still relies on the global variance cutoff percentage hyperparameter, which needs to be set manually.

**Questions:**

- Have the authors studied the computational/memory cost for very large FFN layers (e.g., $n_l > 10000$)? Is there a point where the cost of LDL decomposition itself becomes a practical bottleneck, and have you considered approximate decomposition methods?
- Given the very similar final performance between ZCA and SAW ordering, what is the practical justification for using the more complex ZCA ordering over the simple SAW baseline?
- Have the authors considered a simple hybrid global metric, such as weighting the "cumulative variance" by a first-order Taylor-expansion score, to create a more robust, sensitivity-aware pruning heuristic?
- How did you determine the optimal global variance percentage for your experiments?

---

> ### Author Response · Authors · 2025-11-21
> **Response to Weaknesses**
>
> We thank the reviewer for their positive feedback and questions.
>
> ```The reconstruction provides a "rather small impact" after full retraining, could means that the LLS algorithm is less critical to the final result.```
>
> Indeed, reconstruction seems to be a minor contributing factor after full retraining.
> However, this is also to be expected. Specifically, the reconstruction does not improve the retrained accuracy but instead the immediate accuracy after pruning. This has potential for two particular cases: 1) in cases where one wishes to do no retraining (as is often done for LLMs), or 2) in cases where one wishes to accelerate the retraining process. Notably, our method still reaches competitive performance with many other alternative methods for pruning which we attribute partially to our node importance scoring and partially to this ability for the network to begin retraining from an already somewhat performant state.
>
>
> ```The results are highly competitive but do not universally "match or exceed" SOTA. Its strength is the trade-off, not setting new accuracy records.```
>
> We would like to clarify this concern as it was raised by several reviewers.
>
> Current SOTA methods overwhelmingly use learning-based pruning.
> In the context of SAViT and TPP, the former learns pruning masks, while the latter prunes the network iteratively even before beginning a full retraining pass, an expensive process.
>
> Furthermore, the first step is to applying these methods is often to define compression ratios for different network components (residual blocks (when using TPP), MLP layers, Attention layers, etc.).
> These compression ratios are effectively an additional set of hyperparameters (one per layer) which must trade off accuracy, (theoretical) runtime and memory load.
> How to determine these compression ratios is entirely unaddressed by these existing works.
>
> In contrast, our method is extremely simple, with a single unified variance cutoff for the whole network which automatically translates into layer-wise independent compression ratios.
> We see this as a major benefit of our proposed method.
>
>
> ```The global variance heuristic is effective but limited, as it ignores downstream layer sensitivity, treating all variance as equally important.```
>
> We agree, and we acknowledge this shortcoming in our discussion.
> We believe that a possible solution could involve weighting the global importances by a layer-wise downstream sensitivity score to adjust for this. However, this would involve another significant development of theory which we consider a topic for future work.
>
> ```Have the authors studied the computational/memory cost for very large FFN layers (e.g., )? Is there a point where the cost of LDL decomposition itself becomes a practical bottleneck, and have you considered approximate decomposition methods?```
>
> In the table below, we show that scaling our method to very large layer sizes is feasible, even without approximations. We demonstrate computing our GS reconstruction on increasing layer sizes in single precision and measure the runtime in seconds on an A100 GPU and the peak memory.
>
> Scaling beyond this is still possible via blockwise approximations, but was not considered in this work as scaling to the largest modern networks is still feasible.
>
>  _ | $n_l=8192$	| $n_l=16384$	| $n_l=32768$ | $n_l=65536$
> |--|--|--|--|--
> runtime (s)|	0.9|	2.7|	14.1|	98.8
> peak memory (GB)|	1|	4.1|	16.4|	65.5
>
>
>
> ```Retraining is still necessary. Therefore, it is not a perfect one-shot solution.```
>
> Firstly, we want to clarify that in the literature, one-shot typically means to prune in a single step.
>
> However, indeed, retraining is often necessary when aiming for strong compression.
> Yet, a large body of literature, especially for LLM pruning focuses on pruning without re-training.
> Therefore, we believe that improving cheap reconstruction is still a significant advancement.
>
> Lastly, we believe that with our simple one-step reconstruction we can significantly reduce the amount of re-training time. As we show in Figure 7, our network reaches moderate levels of performance much quicker than no-reconstruction, paving the way for significantly reducing the retraining time.
>
>
> ```While the method automates layer-wise ratios, it still relies on the global variance cutoff percentage hyperparameter, which needs to be set manually.```
>
> We would like to note that compared to alternative work in this space, we have a single hyperparameter (the global variance cutoff) where contemporaries have a compression ration hyperparameter for every layer or layer-type in network's being pruned.
> We have thus already reduced the number of pruning hyperparameters significantly compared to alternative approaches.
> More importantly, our global variance cutoff corresponds to a specific FLOP count reduction. Thus, if one wishes for a specific reduction in FLOPs, a particular global variance cutoff achieves said FLOP reduction and can be determined rather cheaply.

---

> ### Author Response · Authors · 2025-11-21
> **Response to Questions**
>
> ```Given the very similar final performance between ZCA and SAW ordering, what is the practical justification for using the more complex ZCA ordering over the simple SAW baseline?```
>
> Practically, we find that SAW's performance in terms of ordering is not consistently on par with that of a ZCA. Looking at Table 1, one can note that SNP-SAW is consistently of lower accuracy than SNP-ZCA. The only difference between these two methods is indeed a ZCA vs SAW ordering.
>
>
> ```Have the authors considered a simple hybrid global metric, such as weighting the "cumulative variance" by a first-order Taylor-expansion score, to create a more robust, sensitivity-aware pruning heuristic?```
>
> This has been considered but is a significant theoretical departure and practical departure from our current contribution.
> Our current method is focussed on hitting a sweet spot between speed of pruning and final accuracy, considering something like a first-order Taylor-expansion (to weight the variances by sensitivity) could no doubt be powerful but would require the ability to measure gradients via backward passes.
> This costs both computational time, but can also cost significantly in memory usage and it could even be infeasible for extremely large transformer models.
>
>
> ```How did you determine the optimal global variance percentage for your experiments?```
>
> As described above, there is a simple one-to-one correspondence between the variance cutoffs and the compression ratio.
> We have updated our codebase to allow easy computation of the variance cutoff which reproduces a given compression ratio and plan to release the code together with the final version of the manuscript.

---

### Official Review · Reviewer_MsgY · 2025-10-30

**Soundness:** 2
**Presentation:** 2
**Contribution:** 2
**Rating:** 4
**Confidence:** 4

**Summary:**

The authors propose a method of pruning that focuses on pruning away the nodes whose variance can largely be explained by others. They do so via a lower-triangular orthogonalization that implies each subsequent node's "unique" variance residue outlines how much of the node's values can be linearly mapped using the previous ones. As this process is order dependent the authors naturally have to propose a way of optimizing the ordering. This they do via avoiding the permutative computational burden and instead just computing the importance score for each node via its orthogonalization w.r.t remaining nodes. Subsequently, as one still needs to find a consistent strategy for pruning nodes across layers, the authors subsequently appropriately normalize the LDL diagonal elements by computing the cumulative percentage of variance lost if pruned at a certain node. Results are shown on Imagenet, with VGG and Resnet models, and subsequently using Transformer networks and some final no-retraining results for language models.

**Strengths:**

- The authors test their approach on a variety of settings overall, and look at various aspects of the pruning problem such as FLOP and pruning cost.
- Both CNN and Transformer models are investigated.
- Pruning cost is low, as the authors avoid the permutative blow-up of possibilities via fixing the ordering via Importance scoring

**Weaknesses:**

There are some significant issues that come to mind that may need addressing (in no order of importance):

- Although it makes a bit of sense to me why this approach would outperform "without" retraining as the weights are re-parameterized accordingly after pruning away the redundant nodes, as the authors themselves acknowledge re-training is still critical to obtaining good performance. As the main objective of pruning is to reduce compute cost while preserving performance, I don't see the significance of the no-retraining results
- Following up with the point above, I also don't see the immediate significance of the proposed approach having low pruning cost. Ultimately performance matters while reducing the final compute, and the authors are doing re-training to get to the best performance anyway (which adds 10+ hours), so given that much additional time I don't immediately see the utility of low pruning cost in this pipeline. Apart from this, overall, I didn't get the feeling from the results that the improvements, flop and accuracy wise are that significant. Furthermore, I think the comparisons would definitely benefit from more baselines, especially for Table 2.
- A general issue I have with node based pruning is that without the actual impact of each node on the rest of the architecture, it is not clear whether one should prune it. To give an example, consider simple variance based pruning for each node, by ordering the nodes from lowest to highest variance (and then subsequently adapting the layer's bias to account for the node removal). This approach initially seems quite reasonable as nodes with lower variance should have a lower impact on the overall network's decision, after having adjusted for bias. However, it is definitely possible that the network weights have adapted to the lower variance nodes by increasing in magnitude so that the "effective variance" of the low variance nodes may not be low. So in that case, pruning away the lowest variance nodes is not a guarantee that one is actually removing the important ones. And my observation is also that the authors' proposed approach should not be that dissimilar from a simple variance based pruning strategy, as they use "unnormalized-ZCA" ordering, which would have a significant bias towards giving low scores to units with lower variances. So I'm also not sure how different the orderings/pruned nodes would be when compared to simple variance/energy based pruning methods.

**Questions:**

Same as weaknesses.

---

> ### Author Response · Authors · 2025-11-21
>
> We thank the reviewer for their valuable feedback.
>
> ```Although it makes a bit of sense to me why this approach would outperform "without" retraining as the weights are re-parameterized accordingly after pruning away the redundant nodes, as the authors themselves acknowledge re-training is still critical to obtaining good performance. As the main objective of pruning is to reduce compute cost while preserving performance, I don't see the significance of the no-retraining results```
>
>
> We believe that our 'without retraining results' have a few important implications.
>
> Firstly, Figure 5 shows that there is intrinsic redundancy in the networks that is unrelated to the specific input data, but rather an artifact of the model training.
> It shows that if one was to perform importance scoring without considering the network's redundancy that is reconstructable, one severely overestimates the importance of redundant information.
> This speaks to what existing importance measures are perhaps getting wrong.
>
> Secondly, in the LLM pruning literature, models are usually not retrained after pruning. Therefore, the development of methods that accurately prune without requiring retraining are significant.
>
> Lastly, we observe that the increased performance before retraining can allow for more rapid and perhaps more extensive retraining and may also be why we are able to reach competitive levels of accuracy against a range of SOTA methods.
>
>
> ```Following up with the point above, I also don't see the immediate significance of the proposed approach having low pruning cost. Ultimately performance matters while reducing the final compute, and the authors are doing re-training to get to the best performance anyway (which adds 10+ hours), so given that much additional time I don't immediately see the utility of low pruning cost in this pipeline. Apart from this, overall, I didn't get the feeling from the results that the improvements, flop and accuracy wise are that significant. Furthermore, I think the comparisons would definitely benefit from more baselines, especially for Table 2.```
>
> We agree with the latter point, see the updated baselines which we have added to Table 2.
> In response to the former, we would like to contrast our minimal pruning cost against that of TPP.
> The reported time spent on pruning for TPP is 12h and that of retraining is 29h. This amounts to an additional 41% of the retraining time that must be spent on pruning, where our method's pruning time is approximately 1.7% of the retraining time.
> Furthermore, our method is very easy to adapt compared to SOTA approaches. We refer to our response to reviewer Rnnj for a discussion of the complexity to adapt our method.
>
>
> ```A general issue I have with node based pruning is that without the actual impact of each node on the rest of the architecture, it is not clear whether one should prune it. To give an example, consider simple variance based pruning ...```
>
> This concern speaks to the assumptions made in node pruning approaches. Indeed, methods do not always take into account the downstream impact of the metric used as the importance score (in this case the subspace variance). One way to do this more rigorously is to measure node impact on the output loss via a first or second order approximation, though note that these are often expensive to compute. To more properly place our choices into context of the metric being considered and what alternatives might be possible, see Appendix C in the revised manuscript. Please also see our response to reviewer 2DnV for more discussion on this point.

---

### Official Review · Reviewer_2DnV · 2025-10-30

**Soundness:** 3
**Presentation:** 2
**Contribution:** 2
**Rating:** 4
**Confidence:** 2

**Summary:**

This paper proposes Subspace Node Pruning, a method that projects neural activations into an orthogonal subspace to identify and remove redundant units. Pruned nodes are approximately reconstructed through linear least squares, avoiding full retraining. The authors introduce an unnormalized-ZCA-based importance metric and a variance-based global pruning ratio to automate layer-wise sparsity. Experiments on CNNs, Transformers, and OPT language models show comparable or slightly better accuracy than prior work, with lower computational cost.

**Strengths:**

The paper presents a clear and coherent technical idea. Using orthogonal subspaces for pruning is a reasonable formulation, and the derivation based on LDL or Gram-Schmidt decomposition is easy to follow. The proposed global variance criterion provides a practical way to automatically balance pruning ratios across layers without manual tuning. The experimental coverage is fairly good, including CNNs, Transformers, and language models, and the results are generally consistent, indicating that the method is sound and well-implemented.

**Weaknesses:**

(1) The empirical gains over strong baselines appear relatively modest. Improvements are generally small and sometimes not consistent across models or pruning ratios. The results are solid but may not convincingly demonstrate a clear advantage over prior work.

(2) The presentation could be further polished. Some definitions and notations could be clarified, and a few minor typographical or formatting issues slightly affect readability. In addition, the overall structure of the paper is somewhat unconventional, which makes it harder to follow the logical flow and locate key methodological details.

**Questions:**

(1) The use of orthogonalization for pruning appears related to prior work on redundancy-based and subspace reconstruction methods, and it would be helpful to more clearly articulate what is fundamentally new in this formulation and how it differs conceptually from existing techniques.

(2) The claimed efficiency of the method might come with certain trade-offs. In Table 4, even when the amount of data used during pruning is increased, the improvement remains marginal compared with the baselines. The authors are encouraged to clarify whether the reported speed and simplicity come at the cost of performance or data efficiency.

---

> ### Author Response · Authors · 2025-11-21
>
> We thank the reviewer for their valuable feedback and questions.
>
> ```The empirical gains over strong baselines appear relatively modest. Improvements are generally small and sometimes not consistent across models or pruning ratios. The results are solid but may not convincingly demonstrate a clear advantage over prior work.```
>
> This concern was raised by multiple reviewers. In short, the power of this method is particularly in it's simplicity of application and lack of hyperparameter tuning requirement for the compression ratio of every network layer. Given that we have a single global pruning hyperparameter (global variance cutoff) instead of layer-wise compression ratios, the ability for SNP to match SOTA is extremely impressive by our estimation. We refer the reviewer to our response to reviewer Rnnj for a more detailed explanation.
>
> ```The presentation could be further polished. Some definitions and notations could be clarified, and a few minor typographical or formatting issues slightly affect readability. In addition, the overall structure of the paper is somewhat unconventional, which makes it harder to follow the logical flow and locate key methodological details.```
>
> To address this comment, we have thoroughly proof edited the paper and introduced major updates to Section 2.
> Appendix C has also been added to more technically ground the derivation of our method. See the uploaded a revised version of the manuscript. Our major changes are highlighted in purple. We hope these changes address your concern.
>
>
> ```The use of orthogonalization for pruning appears related to prior work on redundancy-based and subspace reconstruction methods, and it would be helpful to more clearly articulate what is fundamentally new in this formulation and how it differs conceptually from existing techniques.```
>
> We refer to the added Appendix C to the manuscript for a comparison to prior work on redundancy based methods. Furthermore, we articulate our fundamental differences in this section, but also adjusted the method section to clearly articulate our contributions.
>
> As a short summary: Several methods have been proposed following Optimal Brain Damage (OBD, LeCun et al. 1989) and Surgeon (OBS, Hassibi and Stork 1992) that minimize the loss change induced by pruning each individual unit.
> While OBD and OBS were introduced in a time where networks were small enough to compute the full Hessian matrix of the network, nowadays, approximations to the Hessian are needed. In response, several approximations have been proposed - most notably by Molchanov et al. (2019) and Wang et al. (2019).
> In these node pruning works, the local importance scores are assumed to be globally calibrated as they approximate a global loss function, thus capturing downstream sensitivity.
> However, when deployed in a one- or few-shot pruning setting, these methods inevitably overestimate each unit's contribution to reconstruction.
> Our formulation of pruning in an orthogonal subspace provides a different perspective and proposes a solution. By separating local and global importance scoring, we adjust the overestimation during global importance scoring with the exact reconstruction possible from each unit under the local ordering.
>
> We present our method with the simplest related importance metric. That is, we choose to measure importance as a layer-wise reconstruction loss within our subspaces. However, both the local and global importance scoring method can be applied to more involved reconstruction losses. This, however, is out of the scope of this work, where we primarily intended to demonstrate the viability of our perspective, as well as the effectiveness of the approach that can be directly derived from it.
>
>
> ```The claimed efficiency of the method might come with certain trade-offs. In Table 4, even when the amount of data used during pruning is increased, the improvement remains marginal compared with the baselines. The authors are encouraged to clarify whether the reported speed and simplicity come at the cost of performance or data efficiency.```
>
> We assume that this concern regards Table 2. Note that the weakness you indicate is in fact a strength that we wished to show: our method can operate well whether using the full dataset or even a fraction of it to compute reconstruction weights. Table 2 also shows that our method, when using the entire training dataset for pruning, is much faster than SAViT. Figures 5 and 10, then additionally show that our method is also effective even when reducing the amount of training data used for reconstruction to just a few mini-batches (~0.1\%). This configuration runs in less than one minute.
>
> This is intended to show that our method scales more favourably compared to the baselines and that minimal data is necessary for a good reconstruction.

---

### Author Response · Authors · 2025-11-29
**Rebuttal Summary**

First, we thank all the reviewers for their in-depth evaluation of our manuscript and are pleased to read that all reviewers see the strength of our global calibration method. We also thank the ACs, especially given the additional effort that is now required for this year.

This comment is intended to summarise the key points of concern from reviewers and how we addressed those, point by point.

1. All reviewers expressed concerns regarding the fact that our method's final model accuracies only match (often marginally beat) SOTA but do not exceed it significantly.

In response, we emphasize that SOTA methods are significantly slower than that which we propose.
Beyond the explicit pruning time, our method also has only a single global pruning hyperparameter (the global variance cutoff) where almost all other methods need extensive hyperparameter tuning (with a pruning ratio parameter often necessary for every layer).
The tuning time required for these extra hyperparamters for other methods is often left unreported, but is significant.
Our method is free of such tuning, and thus our ability to match or marginally outperform SOTA without all such tuning is a significant win.
Furthermore, our method finds applicability in retraining-free pruning cases, such as in applications to LLMs.

2. Reviewers MsgY and Rnnj expressed concerns regarding whether our 'without-retraining' results are useful.

We explain in response, that retaining high accuracy while pruning could, and perhaps should, be one of the most crucial importance scores for pruning.
Our results, therefore, demonstrate that our method cheaply retains significant performance even during pruning and is thus closer to an ideal importance score than other methods.
This is effectively demonstrated in our retraining-free experiments where our networks can remain performant even with significant pruning (but without retraining).

Secondly, we explain that for some applications (e.g. LLM or other huge networks) retraining is prohibitively expensive. In such cases, our method's applicability to re-training free pruning is crucial.
Lastly, we hypothesize that our matching to SOTA may be a result of our network's already starting off with significantly higher performance. This may even allow practitioners to accelerate the retraining process by starting far above chance level.


3. 2DnV and MsgY asked about the similarities and differences between our method and other variance-based pruning methods

In response, we added Appendix C to the manuscript to more technically describe how our method relates to other variance-based measures.
Furthermore, in our response to reviewer 2DvN, we expanded that alternative methods lack a proper calibration to the full set of units being pruned in the one or few-shot setting.
In this work, we propose a novel perspective of pruning within an orthogonal subspace which also operates extremely well in the one-shot regime with extremely low computational cost.

4. Reviewers ARux and 2DnV noted a couple of typos and unclear definitions and assumptions.

In response, we proof edited and made our assumptions and definitions clear in response to reviewers.

5. Reviewer ARux expressed interest in comparisons to SVD-based low rank approximation, inclusion of another dataset, as well as a relation of FLOP savings to runtime.

In our response, we further demonstrate our method by application to ResNet-56 on CIFAR-10 (see Appendix J), a comparison to low-rank approximation by application to ResNet50 and DeiT-tiny on ImageNet (see Appendix K), and a runtime vs FLOPs comparison in Appendix L.
These results consistently show that our method operates efficiently and effectively compared to alternatives.
Namely, our method significantly outcompetes SVD-based low rank approximation in practical terms, has competitive performance on CIFAR-10 in ResNet-56, and that indeed runtime is saved proportionally to our reported FLOP savings.

6. Reviewer Rnnj expressed a concern regarding the scalability of our approach to huge layer-sizes.

In response, we computed the runtime and memory requirements of our method on layers up to 65_536 units in width and showed that our approach remains viable.

---

### Meta-Review · Area_Chair_CYo5 · 2025-12-04

**Summary:**

The key reviewer concerns were:

1. Modest empirical performance
2. A lack of experimental breadth

I do not think these concerns are satisfactorily addressed (please see below) and I do not think any reviewers would have changed their scores (also see below). The scores are 6444 so are leaning towards reject. I propose reject (but not with full confidence) and encourage the authors to run more experiments, and to do a comparison on hyperparameters (and their importance) with competing methods to strengthen this work.

**Reviewer Concerns:**

Reviewer ARuX noted that experimentally this work was lacking in breadth for a pruning paper, and could do with more diverse architectures and tasks, as well as actual throughout speeds. The authors responded to this by adding a CIFAR 10 experiment with a ResNet-56 and added inference time on an A100 for a ResNet-50 and a DeIT-Tiny. I do not think this addresses the concern sufficiently as this is a single extra experiment and benchmarking for two networks. I cannot see from this whether e.g. the networks I get from this method will be faster than for competing methods. Regarding performance, the results are competitive, but not leading. The author’s argue that the appeal is their method only uses a single pruning hyperparameter. What is lacking is analysis showing that all the other methods are different in this respect. If I take, for example, Trainability Preserving Neural Pruning (ICLR 2023) I can see three hyperparameters but two are held constant over 3 datasets (implying their values aren't very sensitive); this does not seem much more elaborate than this approach. What the authors should include is an analysis showing what hyperparameters different methods use and their importance/sensitivity to argue for the appeal of their approach.

**Reviewer Scores:**

I do not think the justification regarding hyperparameters would have swayed Reviewer 2DnV, so I think their score would have remained the same. Reviewer MsgY was (I believe) most concerned about performance and didn’t see the appeal of low pruning cost, and I predict would have retained their score. Reviewer Rnnj is positive leaning; the response addresses most of their concerns but I don’t think provides anything substantial that would push them to a higher positive score. I predict Reviewer ARuX would not have changed their score as the experiments provided didn’t address all that was asked for (e.g. mobile variants of models).

---

### Decision · Program_Chairs · 2026-01-26

Reject